**Seasonal Variation of Total Column Formaldehyde, Nitrogen Dioxide, and Ozone Over Various Pandora Spectrometer Sites with a Comparison of OMI and Diurnally Varying DSCOVR-EPIC Satellite Data**

Jay Herman[1,2] and Jianping Mao[2,3]

[1]GESTAR II University of Maryland Baltimore County, Baltimore, Maryland USA

1000 Hilltop Cir, Baltimore, MD 21250

[2]NASA Goddard Space Flight Center, 8800 Greenbelt Road, Greenbelt, MD 20771, USA

Correspondence: Jay Herman (herman@umbc.edu)

[3]College of Computer, Mathematical and Natural Sciences, University of Maryland, College Park, MD 20740, USA

**Abstract**
Observations of trace gases, $O_3$, HCHO, and $NO_2$, and their seasonal dependence can be observed using
satellite and ground-based data from the Ozone Monitoring Instrument (OMI) satellite and Pandora
ground-based instruments.  Both operate with spectrometers that have similar characteristics in
wavelength range and spectral resolution that enable them to retrieve total column amounts of
formaldehyde (TCHCHO), and nitrogen dioxide (TCNO2), and total column ozone (TCO). The polar
orbiting OMI observes at 13:30 ± 0:25 local time plus an occasional second side-scan point 90 minutes
later at mid-latitudes. The ground-based Pandora spectrometer system observes the direct sun all day
with a temporal resolution of 2 minutes. At most sites, Pandora data show a strong seasonal
dependence for TCO and TCHCHO and less seasonal dependence for TCNO2. Use of a low pass filter
Lowess(3-months) can reveal the seasonal dependence of TCNO2 for both OMI and Pandora at mid-
latitude sites usually correlated with seasonal heating using natural gas or oil. Compared to Pandora,
OMI underestimates the amount of $NO_2$ air-pollution that occurs during most days, since the OMI
TCNO2 retrieval is around 13:30± 0:25 local time, which tends to occur near the frequent minimum of
the daily TCNO2 time series. Even when Pandora data are restricted to between 13:00 and 14:00 hours
local time OMI retrieves less TCNO2 than Pandora over urban sites because of OMI's large field of view.
The seasonal behavior of TCHCHO is mostly caused by the release of HCHO precursors from plant
growth and emissions from lakes that peak in the summer as observed by Pandora and OMI. Long-term
averages show that OMI TCHCHO usually has the same seasonal dependence but differs in magnitude
from the amount measured by Pandora and is frequently larger. Comparisons of OMI total column $NO_2$
and HCHO with Pandora daily time series show both agreement and disagreement at various sites and
days with Pandora frequently larger. For ozone, daily time dependent comparisons of OMI TCO with
those retrieved by Pandora show good agreement in most cases. Additional diurnal comparisons are
shown of Pandora TCO with hourly retrievals during a day from EPIC (Earth Polychromatic Imaging
Camera) spacecraft instrument orbiting the Earth-Sun Lagrange point $L_1$.
**1.0 Introduction**
Formaldehyde, HCHO, is ubiquitous in the atmosphere and as with other VOCs (Volatile Organic
Compounds) are derived from natural and anthropogenic sources, such as plants, animals, biomass
burning, fossil fuel combustion, and industrial processes (Zhang et al., 2019; Morfopoulos et al., 2021).
Formaldehyde is mainly produced from the oxidation of VOCs such as isoprene, methane, and
anthropogenic emissions (Wittrock, 2006). Formaldehyde can also be directly emitted from some
sources, such as vehicle exhaust, tobacco smoke, building materials, and wood burning affecting
pollution levels both indoors and outdoors. The majority of gaseous and atmospheric formaldehyde
derives from microbial and plant decomposition (Peng et al., 2022). HCHO concentrations in the first few
kilometers of the atmosphere vary depending on the location, time of day, season, and meteorological
conditions. Some of the factors that influence total atmospheric column amounts of HCHO are:
• **Solar radiation**: Formaldehyde is photolyzed by solar ultraviolet radiation (Nussbaumer et al., 2021)
and broken down into smaller molecules and radicals. The photolysis rate of formaldehyde depends
on the solar zenith angle, the cloud cover, and the atmospheric composition. Generally,
formaldehyde photolysis is faster in the summer and during midday.
• **Temperature**: The thermal decomposition rate of formaldehyde increases with temperature, which
means it is faster in warmer regions and seasons.
• **Humidity**: Formaldehyde reacts with water vapor in the atmosphere, forming formic acid and
hydroxyl radicals. The reaction rate of formaldehyde with water vapor depends on the relative
humidity, which varies with the temperature and the precipitation. Generally, formaldehyde
reaction with water vapor is faster in humid regions and seasons.

The largest sources of $NO_2$ are obtained from fossil fuel burning from various types of automobiles truck
emissions and power generation followed by industrial processes and oil and gas production (Van der A,
2008; Stavrakou et al. 2020). Additional sources are soils with natural vegetation, oceans, agriculture
with the use of nitrogen rich fertilizers, forest fires, and lightning. In populated areas requiring winter
heating, anthropogenic sources of lower tropospheric $NO_2$ are larger than natural sources. Nitrogen
oxides play a major role in atmospheric chemistry and the production and destruction of ozone in both
the troposphere and stratosphere. In the boundary layer high concentrations of both HCHO (Kim et al.,
2011) and $NO_2$ (Faustini et al., 2014) are health hazards for humans.

TCHCHO, TCNO2 and TCO in the atmosphere are typically measured by satellite and ground-based
instruments.
• Satellite: The Ozone Monitoring Instrument (OMI) is a satellite sensor launched in July 2004 that
measures HCHO, $NO_2$, $O_3$, and other atmospheric constituents from space (Levelt et a. 2018).
Detailed descriptions of the OMI instrument are given in Levelt et al. (2006) and Dobber et al.
(2006). Briefly, OMI is a side scanning spectrometer instrument (270 to 500 nm in steps of 0.5 nm)
with a nadir spatial resolution or 13 x 24 $km^2$. OMI data can be used to monitor their global
distribution and long-term trends, and to investigate the role of $NO_2$ and HCHO in atmospheric
chemistry and air quality (Lamsal et al., 2014; 2015; Boeke et al., 2011). For ozone, DSCOVR (Deep
Space Climate Observatory), located at the Earth-Sun gravitational balance Lagrange point $L_1$,
contains a filter-based instrument EPIC (Earth Polychromatic Imaging Camera) capable of obtaining
TCO once per hour (90 minutes in Northern hemisphere winter) simultaneously for the entire sunlit
globe as the Earth rotates (Herman et al., 2018) with nadir resolution of 18 x 18 $km^2$.
• Ground-based Spectrometer: The Pandora spectrometer system forms a worldwide network of over
150 currently working direct-sun observing instruments that match atmospheric observations with
known laboratory spectra of HCHO, $NO_2$, and $O_3$ to obtain the total vertical column above the
Pandora instrument every 2 minutes from multiple co-added spectra. Pandora uses a single-grating
spectrometer and a charge-coupled device (CCD) 2048 x 64-pixel detector to record the direct-sun
spectra in the ultraviolet and visible wavelength range, 280 – 525 nm with an oversampled 0.6 nm
spectral resolution. The retrieval algorithm is based on a spectral fitting technique to retrieve the
slant column densities of $O_3$, HCHO, $NO_2$ and other gases, and then convert them to vertical column
densities using geometric air mass factors appropriate for direct-sun observations. Pandora
spectrometers have been deployed in various field campaigns and locations to monitor the spatial
and temporal variability of TCHCHO and TCNO2 to validate and improve the satellite observations of
TCHCHO (Herman et al., 2009, Tzortziou et al., 2015, Spinei et al., 2018).

Previous validation studies of TCNO$_2$ and TCHCHO have been made with emphasis on the amount of bias
between ground-based and satellite retrievals of total column $NO_2$ and HCHO (Pinardi et al., 2020; de
Smedt et al., 2021) and references therein. Validation studies using Pandora measurements have shown
that OMI TCNO2 retrievals tend to underestimate the degree of $NO_2$ pollution, especially in urban areas
where the coarse OMI spatial resolution tends to reduce the spatially averaged amount (Celarier et a.,
2008; Lamsal et al., 2014; Judd et al., 2019; Zhao et al., 2019). In addition to the different field of view,
the agreement between OMI and Pandora depends strongly on determining the OMI effective air mass
factor for a wide variety of observing and solar zenith angles (Lorente et al., 2017), whereas Pandora
uses a simple geometric direct sun airmass factor (Herman et al., 2009, Eq 3). Studies of TCHCHO
involving Pandora prior to 2020 are probably not valid because of a problem with internal generation of
HCHO in the Pandora instrument (Spinei et al., 2021). More recent studies (Wang et al. 2022) obtain a
seasonal dependence of surface concentrations similar to the TCHCHO in this study. The largest sources
of error in TCHCHO retrievals are the determinations of the air mass factor for satellite observations and
the fact that ozone and formaldehyde have overlapping absorption spectra so that a small error in
ozone retrieval can affect the formaldehyde results. A comparison of direct-sun Pandora TCHCHO
retrievals with Geostationary Environment Monitoring Spectrometer GEMS shows a similar seasonal
dependence (Fu et al., 2025).

Table 1 List of 30 Pandora locations used in this study in order of appearance

| | Pandora Number | Pandora location name | Lat (deg) | Long (deg) | Alt(m) |
|---|---|---|---|---|---|
| 1 | Pan 180 Fig.1,2 | Bronx, New York USA | 40.868 | -73.878 | 31 |
| 2 | Pan 64 Fig.3 | New Haven, Connecticut USA | 41.301 | -72.903 | 4 |
| 3 | Pan 190 Fig.4 | Bangkok, Indonesia | 13.785 | 100.540 | 6 |
| 4 | Pan 182 Fig.5 | Tel Aviv, Israel | 32.113 | 34.806 | 8 |
| 5 | Pan 159 Fig. 6 | Wakkerstroom, South Africa | -27.349 | 30.144 | 18 |
| 6 | Pan 20 Fig.7 | Busan, Korea | 50.798 | 4.358 | 107 |
| 7 | Pan 145 Fig.10 | Toronto-Scarborough, Canada | 43.784 | -79.187 | 14 |
| 8 | Pan 134 Fig. 12 | Bristol, Pa, USA | 40.107 | -74.882 | 10 |
| 9 | Pan 204 Fig. 12 | Boulder, Co USA | 40.038 | -105.242 | 161 |
| 10 | Pan 106 Fig.12,A2 | Innsbruck, Austria | 47.264 | 11.385 | 616 |
| 11 | Pan 117 Fig.12 | Rome Italy | 41.907 | 12.5158 | 75 |
| 12 | Pan 193 Fig.12 | Tsukuba, Japan | 36.066 | 140.124 | 51 |
| 13 | Pan 140 Fig.13,A2 | Washington, DC USA | 38.922 | -77.012 | 6 |
| 14 | Pan 166 Fig.7,A2 | Philadelphia, Pa  USA | 39.992 | -75.081 | 6 |
| 15 | Pan 238 Fig.14 | Granada | 37.164 | -3.605 | 7 |
| 16 | Pan 240 Fig. 14 | Thessaloniki, Greece | 40.6336 | 22.9561 | 60 |
| 17 | Pan 66 Fig.15 | Huntsville Alabama USA | 34.725 | -86.646 | 22 |
| 18 | Pan 156 Fig.15 | Hampton, Virginia USA | 37.020 | -76.337 | 19 |
| 19 | Pan 39 Figs.12,15 | Dearborn, Michigan USA | 42.307 | -83.149 | 18 |
| 20 | Pan 101 Fig.A1 | Izania, Spain | 28.309 | -16.499 | 24 |
| 21 | Pan 119 Fig.A1,A2 | Athens, Greece | 37.998 | 23.775 | 130 |
| 22 | Pan 124 Fig.A1 | Comodoro Rivadavia | -45.7833 | -67.45 | 46 |
| 23 | Pan 131 Fig. A1 | Palau | 7.3420 | 134.4722 | 23 |
| 24 | Pan 135 Fig.A1,A2 | CCNY Manhattan NY USA | 40.815 | -73.951 | 34 |
| 25 | Pan 142 Fig.A1 | Mexico City, Mexico | 19.326 | -99.176 | 2280 |
| 26 | Pan 146 Fig.A1 | Yokosuka, Japan | 35.321 | 139.651 | 5 |
| 27 | Pan 147 Fig.A1 | Detroit, Mi USA | 42.303 | -83.107 | 178 |
| 28 | Pan 150 Fig.A1,A2 | Ulsan, Korea | 35.575 | 129.190 | 38 |
| 29 | Pan 154 Fig.A1 | Salt Lake City Ut, USA | 40.766 | -75,081 | 1455 |
| 30 | Pan 162 Fig.A1 | Brussels, Belgium | 50.798 | 4.358 | 107 |


This study will examine the offsets and seasonal cycles of total column $NO_2$, HCHO, and $O_3$ seen by the
Pandora instruments by examining multi-year (2021 – 2024) time series for seasonal and daily behavior
at various sites and will compare with observations made from the OMI satellite overpass
measurements (based on OMI gridded $0.25^O$ x $0.25^O$ data) for the Pandora sites. Pandora ozone
measurements will be additionally compared to hourly data obtained from EPIC. All of the Pandora data
used in this study are after the upgrade of the instruments to eliminate internal sources of HCHO
(Spinei, et al., 2021). Part of this study (TCNO2 and TCO) is an extension of Herman et al. (2019) using
Pandora data (2012 – 2017) before the internal upgrade. A difference is that Pandora TCO is now
compared with hourly TCO retrieved by DSCOVR-EPIC. Table 1 shows a list of 30 Pandora sites used in
this study.

**2.0 Examples of Seasonal and Daily Variation of HCHO and NO$_2$**
Worldwide Pandora total column data can be downloaded from the Austrian Pandonia project website
https://data.pandonia-global-network.org/. Of interest for this study are the Level-2 (L2) time series
ASCII files for direct-sun observations. For example, the Bronx New York City files for Pandora
instrument 180 for TCNO2 data are in Pandora180s1_BronxNY_L2_**rnvs3p1-8**.txt, TCHCHO in
Pandora180s1_BronxNY_L2_**rfus5p1-8**.txt, and TCO data in Pandora180s1_BronxNY_L2_**rout2p1-8**.txt
with the 9 bold characters identifying the file contents. This naming convention applies to all Pandora
sites.
The Pandora data are arranged in irregular columns that are identified in the metadata header for each
file. In the current version, column 1 contains the GMT date and time for each measurement and
column 39 contains measured column density in moles m$^{-2}$ (multiply by $6.02214076 \times 10^{23}/2.6867 \times 10^{20}$ =
2241.4638 to convert to DU where 1 DU = $2.6867 \times 10^{20}$ molecules m$^{-2}$). Pandora data also contain
measurements of water vapor, and SO$_2$ total column amounts in different files.

The original OMI data has a resolution of 13 x 24 km$^2$ at the center of the OMI side-to-side scan. The
overpass OMI data is based on the latest gridded version with 0.25$^O$ x 0.25$^O$ pixel resolution
(midlatitudes approximately 30 x 30 km$^2$). The closest OMI pixel to each Pandora site within 50 km
is used for time-matched comparisons. Long time series use all available Pandora data between
07:00 and 17:00 filtered for data quality (values with large RMS errors and with negative values are
removed). Diurnal comparisons with OMI on specified days use Pandora minute-by-minute data
that are nearly continuous suggesting that Pandora is observing the direct sun under clear-sky
conditions. Clouds cause some scatter in consecutive data points.
Figure 1 shows the seasonal and daily variation of total column HCHO (TCHCHO) and NO$_2$ (TCNO2) in
Bronx, New York. The daily data for 1 week in July and September shows the range of values for both
weekdays and weekends. When all the Bronx TCHCHO data are plotted as an aggregate for 3 years, there
is a strong seasonal pattern with a maximum in July and a minimum near the end of December. The
summer seasonal dependence of TCHCO is consistent with the surface HCHO values observed by the
ground-based Air-Quality System AQS (Wang et al., 2022).  For TCNO2, there is a weaker seasonal
pattern as shown in the Lowess(0.033) fit to the data (Cleveland, 1979; Cleveland and Devlin, 1988) with
moderate maxima in January-February, since the sources of NO$_2$ are largely from the nearly constant
flow of cars and trucks. The parameter 0.033 is the fraction of the time-series data included in the local
least squares estimate, or about 1 month for Pan 180.

Figure 2 shows the daily average of Pandora data obtained from diurnal variation of TCHCHO and $TCNO_2$
from 09:00 to 15:00 local standard time (GMT – 5). The primary emission sources of atmospheric HCHO
include direct emissions of HCHO precursors from vegetation and lakes, primarily through the release of
biogenic volatile organic compounds such as isoprene and terpenes from vegetation, the soil, biomass
burning, and decaying plant and animal matter. This is consistent with the Bronx location that is
adjacent to a large, vegetated park with a small lake near Fordham University. The same TCHCHO
seasonal dependence and magnitude occurs when the Pandora sampling is restricted to 13:00 to 14:00
local standard time similar to the OMI overpass time.
There are 3 Pandora sites in New York City and one in nearby Bayonne, New Jersey. The NYC sites are in
the Bronx-Fordham University, Manhattan-City College NY (CCNY), Queens-Queens College. All four
successfully measured $NO_2$ in the period 2021 – 2023. A strong seasonal cycle in TCNO2 is not seen (Figs.
1 and 2) in the traffic driven production of $NO_2$.in the Bronx, New York. The mean values of total column
$NO_2$ (TCNO2) for each of the 3 New York sites are 0.5 DU while the TCNO2 for the port city of Bayonne,
NJ is substantially higher at 0.7 DU. None of the four sites show a large seasonal daily average TCNO2
pattern. For TCHCHO, all four sites show an annual seasonal cycle with three of the sites having a 3-year
average of 0.3 DU except for the Queens site at 0.45 DU. The Queens site may be anomalous because of
many missing points affecting the average.
Similar behavior is seen at other sites such as the one from New Haven Connecticut located in a
vegetated area adjacent to two rivers (Fig.3). TCHCHO has a clear summer peak in June – July and a
weak winter TCNO2 peak in December to January coinciding with the maximum heating season.
The seasonal variation of TCHCHO could not be studied prior to the internal upgrade of Pandora after
2019 that was needed because of the release of HCHO from polyoxymethylene (POM-H Delrin) out-
gassing as a function of daytime temperature within the Pandora sun-pointing optical head (Spinei et al.,
199    2021)

An equatorial Pandora site (Fig. 4) with a sufficiently long data record is located in Bangkok, Indonesia
near a small park and lake. Bangkok has a tropical monsoon climate with three main seasons: hot season
from March to June, rainy season from July to October, and cool season between November and
February. TCHCHO has a seasonal cycle peaking in March – April when the sun is nearly overhead and a
minimum during the rainy season. TCNO2 has a clear seasonal cycle peaking in December – January and
a minimum during the rainy season. Bangkok has a tropical climate with April as the hottest month with
temperatures averaging at 30.5 °C (87°F) and the coldest is December at 26 °C (79°F).

An unusual counter example to the typical TCHCHO seasonal cycle is for the Pandora site located in Tel
Aviv Israel. Tel Aviv has significant amounts of HCHO but does not show seasonal variation in TCHCHO
because of a coastal location in a warm climate even at midlatitudes located at 32.113$^O$N, 34.085$^O$E that
has essentially two seasons, a cool, rainy winter: October – April and a dry, hot summer: May –
September. The result is there is a limited seasonal increase in vegetational activity and almost no
seasonal variation in HCHO (Fig. 5). However, TCNO2 shows a clear seasonal increase in the December -
January months frequently reaching over 0.5DU. The TCNO2 seasonality is similar to that of the near-
surface concentrations reported by Boersma et al., (2009). The Pandora instrument 182 is located at Tel
Aviv University about 1 km from a major highway. Tel Aviv has frequent episodes of smog associated
with heavy automobile and truck traffic (Newmark, 2001). Heating and cooling in Tel Aviv are mainly
electrical with the maximum power generation occurring in the summer, suggesting that the winter
TCNO2 peak is not caused just by electrical power generation from natural gas that emits $NO_2$.
Finally, a Pandora example from the Southern Hemisphere SH from Wakkerstroom, South Africa located
in a rural area near the ocean a few degrees outside of the equatorial zone at $-27.359^OS$ and $30.144^OE$.
As expected, the peak value of TCHCHO occurs near the SH summer in November – December, while
TCNO2 has no significant seasonal dependence.
**2.1 Comparisons Between Pandora and OMI Retrievals of $NO_2$ and HCHO**
In this section three types of comparisons of Pandora with OMI satellite data are considered. First (Fig. 7
upper panels), is the TCNO2 time series consisting of the data record of Pandora and OMI from 2020 –
2023. The second (Fig. 7 lower panels) is a low-pass Lowess(3-months) filter of midday TCNO2 showing
the seasonal variation. The third (Fig. 8), looks at a few selected days in May, July, and December and
compares typical Pandora clear-sky values with the mid-afternoon OMI overpass at times near 13:30
hours equator crossing time. Pandora and OMI data are matched at the same GMT and then converted
to local solar time, GMT + Longitude/15. The OMI overpass HCHO, $NO_2$ and $O_3$ data, 2004 – 2025, are
found at https://zenodo.org/uploads/15468213 in 7Zip ASCII format.
Figure 7 (upper 2 panels) illustrates that OMI only captures the mid-day fraction of the daily values of
total column $NO_2$ and fails to detect the extent of the daily pollution at both the Bronx New York City
and Busan Korea sites. This is because OMI and other polar orbiting satellites only collect data once per
day (occasionally twice per day) at any given location at mid-afternoon, frequently when TCNO2 is
below its daily maximum (Lamsal et al., 2015; Herman et al., 2019).  The lower 4 panels of Fig. 7 reveal
the seasonal dependence of TCNO2 at two mid-latitude Northern Hemisphere sites found by using a 3-
month low-pass filter Lowess(3 Months) showing that there is an annual TCNO2 cycle peaking in the
winter that corresponds to the natural gas and oil heating use.  The Pandora (13:00 to 14:00) values are
larger than those from OMI especially at Busan suggesting that the OMI gridded overpass field of view
$0.25^O$ x $0.25^O$ includes areas of lower $NO_2$ values over the nearby ocean. In the case of the Bronx, the
differences are smaller but also include areas over rivers. Philadelphia Pennsylvania is landlocked but
smaller than an OMI gridded footprint so that the OMI field of view contains somewhat less polluted
suburbs making the OMI TCNO2 closer to the Pandora values. The Boulder Colorado Pandora is in a
small landlocked city where the OMI field of view extends over sparsely populated regions leading to
OMI TCNO2 lower than Pandora values.
Figures 8 and 9 show the diurnal daytime variation for 3 selected days for Pandora retrieved total
column $NO_2$ and HCHO compared with OMI at the overpass time for both the Bronx in New York City
Busan, Korea and Philadelphia, Pennsylvania. These are typical examples of the highly variable hourly
variation of TCHCHO and TCNO2 as observed by Pandora on clear-sky days at most sites.
The hourly variation of TCHCHO and TCNO2 on any given day can take on unique shapes depending on
the presence of surface winds, changes in temperature, and the amount of sunlight. The variability of
TCNO2 is also driven by the strength of the sources (automobile exhaust, power generation, industry,
etc.) as well as the meteorological conditions. On some days, there is good agreement (within 10%) but
in general the OMI overpass values do not agree with Pandora retrieved values for both TCHCHO and
TCNO2. In the sample shown in Figures 8 and 9, the cases of agreement are about 70% of the time for
TCNO2 and 30% for TCHCHO. Also, the OMI TCNO2 frequently is less than the daily maximum of TCNO2.
Figure 9 illustrates the comparison of TCHCHO retrievals from Pandora and OMI. The spectral fitting
algorithm for detecting HCHO absorption is in the same short wavelength UV spectral region as used for
ozone retrieval, 300 – 360 nm (Gratien et al. 2007). This means that the retrieval sensitivity for "seeing"
all the way to the surface is reduced because of ozone absorption and Rayleigh scattering. Also, small
errors in ozone retrieval can affect the detection of HCHO. This problem is not present for the spectral
fitting of $NO_2$, since that usually occurs in the visible range 410 – 450 nm where there is only
interference from a weak and narrow water vapor line.
Pandora TCHCHO daily average data (Fig. 10) for University of Toronto in Toronto-Scarborough (Lat =
43.784$^O$N, Lon = -79.187$^O$W) shows clear peaks in the summer from the vegetation in a surrounding park
area whereas TCNO2 shows only small seasonal variation with small peaks also occurring in the summer
for values less than 0.4 DU. Higher values do not show any seasonal variation. The University of Toronto
is located near a major highway, which is a strong source of $NO_2$ from automobiles and trucks. Unlike
many sites, OMI TCHCHO data over Toronto East (centered on 43.74$^0$N, -79.27$^0$E is about 8 km from the
Pandora site) also shows sporadic summer peak values that are higher than the Pandora 13:00-14:00
averages and all of the Pandora data (Fig. 11).
Using the daily average Pandora data over Toronto-Scarborough (Fig. 10 upper right) shows no visible
hint of an TCNO2 annual cycle that peaks in winter while the OMI TCNO2 amounts at 13:40 show a clear
peak in December – January corresponding to the peak winter heating for the city (Figs. 10 lower right).
Instead of the daily average data, using the average TCNO2 from 13:00 to 14:00 to correspond to the
OMI overpass time and then applying a Lowess(3 month) low-pass filter (Fig. 11) shows less TCNO2 and a
weaker annual cycle  that corresponds to the annual cycle observed by OMI. The OMI FOV includes the
city of Toronto.

The lower panel in Fig. 11 reproduces the inset values showing the OMI has a stronger TCNO2 annual
cycle because it includes the city area of Toronto. Pandora 145 picks up a small amount of the seasonal
signal from Toronto.
As shown in Fig. 12, the TCHCHO low-pass filtered time series (2021 – 2024), Lowess(3-months),
measured by OMI and Pandora frequently do not agree. An example is the comparison over Bronx, NY
(Lat = 40.868$^O$   Lon = -73.878 $^O$) where the Pandora 180 is located in a park with a small lake, while OMI
gridded data is averaged over a large area 33 x 33 $km^2$ in New York City with little vegetation. In 5 of the
6 sites shown in Fig.12, the OMI retrieval shows more TCHCHO than observed by Pandora. Tsukuba,
Japan is an exception. Six additional sites are shown in the Appendix Fig. A2.
The disagreement over Boulder Colorado may be caused by OMI's large field of view that includes lower
altitude grasslands. Similarly, the Innsbruck Pandora is located in a valley at the University of Innsbruck
surrounded by mountain areas where TCHCHO varies over the OMI FOV. Except for a few cases (e.g.,
Bronx, NY and Innsbruck, Austria) OMI and Pandora see the same TCHCHO annual cycle.


**2.2 Total Ozone Column**

The retrieval of total column ozone amounts TCO (Figs. 13) serves as a check on the calibration of both
OMI and Pandora that is also needed for spectrally overlapping TCHCHO retrievals. Comparisons of
Pandora TCO with TCO measured by OMI show good agreement suggesting both instruments are well
calibrated in the UV range also needed for retrieving TCHCHO. The good TCO agreement is partly
because most of the $O_3$ is in the stratosphere near 25 km and the fact that ozone is slowly changing
spatially over the OMI field of regard for the overpass data. Figure 13 shows an example obtained over
Washington DC from the roof of the NASA Headquarters building and from the roof of a building at
Pusan University, Korea. The other sites in Table 1 show similar good monthly average agreement.

A test of Pandora UV data is a comparison between EPIC, OMI and Pandora TCO at the specific OMI and
EPIC overpass times (Fig. 14 and 15). that shows good agreement within 1 to 3 %. OMI TCO overpass
data for all Pandora sites and more are available from https://zenodo.org/uploads/15468213 in 7Zip
ASCII format. There is also good agreement between daily OMI TCO with that obtained from Pandora
(Fig. 14) at most sites. The values obtained at Granada differ by about 8 DU or 2.9 %.

The diurnal variation of TCO seen by Pandora can be compared (Fig. 15) with that observed by the Earth
Polychromatic Imaging Camera (EPIC) on the DSCOVR (Deep Space Climate Observatory) satellite
orbiting about the Earth-Sun gravitational balance Lagrange-1 point (Herman et al., 2018). EPIC obtains
simultaneous data from sunrise to sunset once per hour (once per 90 minutes during Northern
Hemisphere winter) as the Earth rotates in EPIC's FOV (field of view). Examples of EPIC's view of the
whole illuminated Earth are available from https://epic.gsfc.nasa.gov/.  The spatial resolution for TCO is
18 x 18 $km^2$ at the center of the image (the color images have 10 x 10 $km^2$ resolution). Retrievals earlier
than 07:00 and after 17:00 are not reliable for EPIC or Pandora because of high solar zenith angle effects
(spherical geometry effects for SZA > 75$^O$) not included in the retrieval algorithms. In the case of EPIC,
this is compounded by high View Zenith Angles VZA outside of 07:00 to 17:00 local sun time.
For the cases shown, the TCO data are properly retrieved between 07:00 and 17:00 local solar time. The
10:20 and 11:30 EPIC value for Hampton, VA of 286.5 and 285DU differs from Pandora by -3 %. Other
differences are smaller. Occasionally, OMI differs from Pandora values as is the case, -4.6 %, for 21
August 2023 over Washington, DC.

**3.0 Summary**

Typical examples of the seasonal variability of HCHO, $NO_2$, and $O_3$ in terms of their measured total
column TCHCHO, TCNO2, and TCO have been presented from both ground-based Pandora Spectrometer
instruments and the OMI satellite spectrometer instrument overpass retrievals for selected Pandora
sites. For most sites, OMI observes the strong seasonal variation of TCHCHO that is also clearly seen in
the Pandora data and in surface measurements (Wang et al., 2022).  OMI TCHCHO retrievals are usually
larger than those retrieved by Pandora but not always (Fig. A2). The amount of seasonal variation for
TCHCHO varies depending on the site. For most midlatitude sites, the seasonal variation is significant
with peak values occurring during the summer.

A comparison between the multi-year time series of Pandora and OMI TCNO2 in urban areas shows that
OMI is underestimating the degree of atmospheric $NO_2$ pollution. The results for TCNO2 and TCO agree
with Pandora data, 2012 – 2017, from a previous study before the Pandora upgrade (Herman et al.,
2019). When Pandora is limited to an average of data obtained between 13:00 and 14:00 hours, the
agreement between Pandora and OMI TCNO2 is better but with OMI TCNO2 frequently less than
observed by Pandora.  Comparisons of Pandora daily diurnal time series of TCHCHO and TCNO2 with
OMI overpass values show agreement about 30% and 50 % of the time, respectively with OMI frequently
retrieving more TCHCHO than Pandora.
OMI TCNO2 at one shown site, Toronto-Scarborough, shows seasonal variability that the Pandora 145
does not appear to see. However, limiting the data to the OMI overpass time between 13:00 and 14:00
and applying a Lowess(3-months) low-pass filter reveals a weak annual cycle compared to OMI. This
could be because OMI detects the $NO_2$ source from winter heating in the city, while the Pandora site
(University of Toronto campus) is fairly remote from Toronto city buildings and is mostly affected by road
traffic as the source of $NO_2$. The same low-pass filter technique applied to other sites (e.g., Bronx, NY,
Busan, Korea, Philadelphia, Pennsylvania, and Boulder, Colorado) also show an annual cycle
corresponding to winter heating based on combustion.
Total column ozone agrees well in both seasonal variation and in comparison with Pandora at the OMI
overpass time. Given the nature of the ozone retrieval algorithm, the good agreement with TCO suggests
that the UV calibrations for the Pandoras and OMI are correct. At most well-calibrated Pandora sites,
there is good agreement between Pandora TCO with the hourly TCO obtained from the DSCOVR-EPIC
instrument observing the Earth from an orbit about the Earth-Sun gravitational balance Lagrange-1
point.





**Appendix**

Figure A1 shows the seasonal dependence of TCHCHO with the majority of sites showing a maximum
TCHCHO in mid-summer.

Figure A2 shows additional cases where OMI and Pandora see the same seasonal dependence but differ
on the amount of TCHCHO retrieved.

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

**Author contribution**:
JH is responsible for writing the paper and creating the figures. JM obtained the EPIC overpass data
for the Pandora sites and discussed aspects of the paper.
**Data Availability**
Worldwide Pandora data for 63 sites is available from the Austrian Pandonia project website
https://data.pandonia-global-network.org/
The OMI overpass HCHO, $NO_2$ and $O_3$ data, 2004 – 2025, are found at
https://zenodo.org/uploads/15468213 in 7Zip ASCII format.

**Competing interests**:
The authors declare that they have no conflicts of interest.

Funding: This study is funded by the DSCOVR-EPIC project through the University Of Maryland
Baltimore County

**Acknowledgements**:
The authors want to acknowledge the contribution of each of the Pandora Principal Investigators
included in the figure captions and for the OMI team and Dr. Lok Lamsal for making OMI overpass
data available. Acknowledgement is also due to the Pandonia team lead by Dr. Alexander Cede for
processing all of the Pandora data and devising the retrieval algorithms and to Dr. Nader Abuhassan
for building and calibrating all of the Pandora spectrometer systems. The Pandonia Global Network
PGN is a bilateral project supported with funding from NASA and ESA.

**Figure Captions**
Fig. 1 Seasonal and daily behavior of HCHO and $NO_2$ from Pan 180 located in the Bronx, NYC at $40.868^O$N,
$-73.878^O$W. The blue lines are a Lowess(0.033) fit to the data (light grey), which is approximately a 1-
month local least-squares average. The Local principal investigator for Pan 180 is Dr. Luke Valin.
Fig. 2 The daily average seasonal variation of HCHO and $NO_2$ over Fordham University in Bronx, New
York City from Pandora 180 at $40.868^O$ latitude, $-73.878^O$ longitude, and 0.003 km altitude. Each point is
a daily average of the data in Fig.1. Local principal investigator: Dr. Luke Valin
Fig. 3 The seasonal variation of TCHCHO and TCNO2 over New Haven Connecticut from Pandora 64 at
$41.301^O$N latitude and $-72.903^O$W longitude. Each point is a daily average. Local principal investigator:
Dr. Nader Abuhassan.
Fig. 4 The seasonal variation of TCHCHO and TCNO2 over equatorial Bangkok Indonesia at $13.785^O$N and
$100.540^O$E. The local principal investigator is Surassawadee Phoompanit.
Fig. 5 Seasonal variation in daily average TCHCHO and TCNO2 in Tel Aviv Israel from Pandora 182 located
at $32.113^O$N $34.085^O$E at a height of 8 meters. The local principal investigator for Pan 182 is Dr. Michal
Rozenhaimer.
Fig. 6 Seasonal variation in daily average HCHO and $NO_2$ in Wakkerstroom South Africa from Pandora
159 located at $-27.359^O$S and $30.144^O$E. Local principal investigator: B. Scholes
Fig. 7 Upper 2 Panels: Comparison of OMI (approximately 13:30) and Pandora (07:00 – 17:00) total
column $NO_2$ time series in Bronx NY ($40.868^O$N, $-73.878^O$W) and Busan Korea ($35.235^O$N, $129.083^O$E).
Lower 4 Panels: Pandora data for Bronx, Busan, Philadelphia ($39.992^O$N, $-75.081^O$W) and Boulder
($40.0375^O$N, $-105.242^O$W) are averaged between 13:00 – 14:00 hours. Both OMI (blue) and Pandora
(black) then have a Lowess(3-month) low-pass filter applied. Local principal investigator for Pan20 is Jae
Hwan Kim, for Pan 180 and Pan 166 is Dr. Luke Valin, and for Pan 204 Dr. Nader Abuhassan.
Fig. 8 A comparison between Pandora and OMI (Orange circle) total column $NO_2$ for 3 locations (Bronx,
New York, Busan Korea, Philadelphia, Pennsylvania. The Local principal investigator for Pan 180 and Pan
166 is Dr. Lukas Valin and for Pan 20 is Dr. Jae Hwan Kim.
Fig. 9 A comparison between Pandora and OMI (purple circle) total column HCHO. The Local principal
investigator for Pan 180 is Dr. Luke Valin and for Pan 20 is Dr. Jae Hwan Kim.
Fig. 10 A comparison of Pandora TCHCHO and TCNO2 daily average total column amounts for Toronto-
Scarborough University of Toronto and OMI data for Toronto East ($43.740^O$N, $-79.270^O$W at
approximately 13:20±0:20 Local Sun Time, GMT + Longitude/15). The local principal investigator for Pan
145 is Dr. Vitali Fioletov.
Fig. 11 TCNO2 annual cycle for Toronto Scarborough from Pan 145 average between 13:00 and 14:00
and OMI. The smooth curves are Lowess(6 Months).

Fig. 12 A comparison between low-pass filtered, Lowess(3 months), OMI and Pandora at six sites with

varying degrees of agreement with TCHCHO(Pan) < TCHCHO(OMI). The Local Principal Investigators are P106 Dr. Stefano Casadio, Dr. Kei Shiomi P193, Dr. Alexander Cede P204, Dr. Lukas Valin P39; P134, and
Dr. Martin Tiefengraber P106. Latitudes and longitudes are in each upper left corner.
Fig. 13 A comparison of OMI Total Column Ozone values with those obtained from Pandora 140 over the
Washington DC site at 38.922$^O$N and -77.012$^O$W and with those obtained from Pandora 20 over the
Busan, Korea site at 32.325$^O$N and 129.083$^O$E. The smooth curves (right panel) are Lowess(6-month) fits
to data in the left panel. The local principal investigator for Pan 140 is Dr. Jim Szykman and for Pan20 is
Jae Hwan Kim.
Fig. 14 A comparison of Pandora and OMI retrievals of total column $O_3$ at the time of the OMI satellite
overpass. Local Principal Investigators: Pan 240 Alexander Cede, Pan 238 Inmaculada Foyo Moreno, Pan
166 Lukas Valin, and Pan 190 Surassawadee Phoompan.
Fig. 15 A comparison of Pandora (Open Circles), EPIC (magenta stars), and OMI (orange circles) retrievals
of total column $O_3$ at the times of the satellite overpasses. Local Principal Investigators: Pan 145 Vitali
Fioletov, Pan 66  Lukas Valin, Pan 39 Lukas Valin, Pan 156 Alexander Cede, Pan 66 Nader Abuhassan,
Pan180 Lukas Valin, and Pan 140 Jim Szykman.
Fig. A1 The seasonal cycle of TCHCHO in DU from 20 randomly selected Pandora TCHCHO time series.
The numbers in the upper left corner are the latitude and longitude in degrees and the Pandora
instrument number in the right corner.
Figure A2 shows additional cases where OMI and Pandora see the same seasonal dependence but differ
on the amount of TCHCHO retrieved.

**Tables**

Table 1 List of 30 Pandora locations used in this study in order of appearance

| | Pandora Number | Pandora location name | Lat (deg) | Long (deg) | Alt(m) |
|---|---|---|---|---|---|
| 1 | Pan 180 Fig.1,2 | Bronx, New York USA | 40.868 | -73.878 | 31 |
| 2 | Pan 64 Fig.3 | New Haven, Connecticut USA | 41.301 | -72.903 | 4 |
| 3 | Pan 190 Fig.4 | Bangkok, Indonesia | 13.785 | 100.540 | 6 |
| 4 | Pan 182 Fig.5 | Tel Aviv, Israel | 32.113 | 34.806 | 8 |
| 5 | Pan 159 Fig. 6 | Wakkerstroom, South Africa | -27.349 | 30.144 | 18 |
| 6 | Pan 20 Fig.7 | Busan, Korea | 50.798 | 4.358 | 107 |
| 7 | Pan 145 Fig.10 | Toronto-Scarborough, Canada | 43.784 | -79.187 | 14 |
| 8 | Pan 134 Fig. 12 | Bristol, Pa, USA | 40.107 | -74.882 | 10 |
| 9 | Pan 204 Fig. 12 | Boulder, Co USA | 40.038 | -105.242 | 161 |
| 10 | Pan 106 Fig.12,A2 | Innsbruck, Austria | 47.264 | 11.385 | 616 |
| 11 | Pan 117 Fig.12 | Rome Italy | 41.907 | 12.5158 | 75 |
| 12 | Pan 193 Fig.12 | Tsukuba, Japan | 36.066 | 140.124 | 51 |
| 13 | Pan 140 Fig.13,A2 | Washington, DC USA | 38.922 | -77.012 | 6 |
| 14 | Pan 166 Fig.7,A2 | Philadelphia, Pa  USA | 39.992 | -75.081 | 6 |
| 15 | Pan 238 Fig.14 | Granada | 37.164 | -3.605 | 7 |
| 16 | Pan 240 Fig. 14 | Thessaloniki, Greece | 40.6336 | 22.9561 | 60 |
| 17 | Pan 66 Fig.15 | Huntsville Alabama USA | 34.725 | -86.646 | 22 |
| 18 | Pan 156 Fig.15 | Hampton, Virginia USA | 37.020 | -76.337 | 19 |
| 19 | Pan 39 Figs.12,15 | Dearborn, Michigan USA | 42.307 | -83.149 | 18 |
| 20 | Pan 101 Fig.A1 | Izania, Spain | 28.309 | -16.499 | 24 |
| 21 | Pan 119 Fig.A1,A2 | Athens, Greece | 37.998 | 23.775 | 130 |
| 22 | Pan 124 Fig.A1 | Comodoro Rivadavia | -45.7833 | -67.45 | 46 |
| 23 | Pan 131 Fig. A1 | Palau | 7.3420 | 134.4722 | 23 |
| 24 | Pan 135 Fig.A1,A2 | CCNY Manhattan NY USA | 40.815 | -73.951 | 34 |
| 25 | Pan 142 Fig.A1 | Mexico City, Mexico | 19.326 | -99.176 | 2280 |
| 26 | Pan 146 Fig.A1 | Yokosuka, Japan | 35.321 | 139.651 | 5 |
| 27 | Pan 147 Fig.A1 | Detroit, Mi USA | 42.303 | -83.107 | 178 |
| 28 | Pan 150 Fig.A1,A2 | Ulsan, Korea | 35.575 | 129.190 | 38 |
| 29 | Pan 154 Fig.A1 | Salt Lake City Ut, USA | 40.766 | -75,081 | 1455 |
| 30 | Pan 162 Fig.A1 | Brussels, Belgium | 50.798 | 4.358 | 107 |



**Figures**

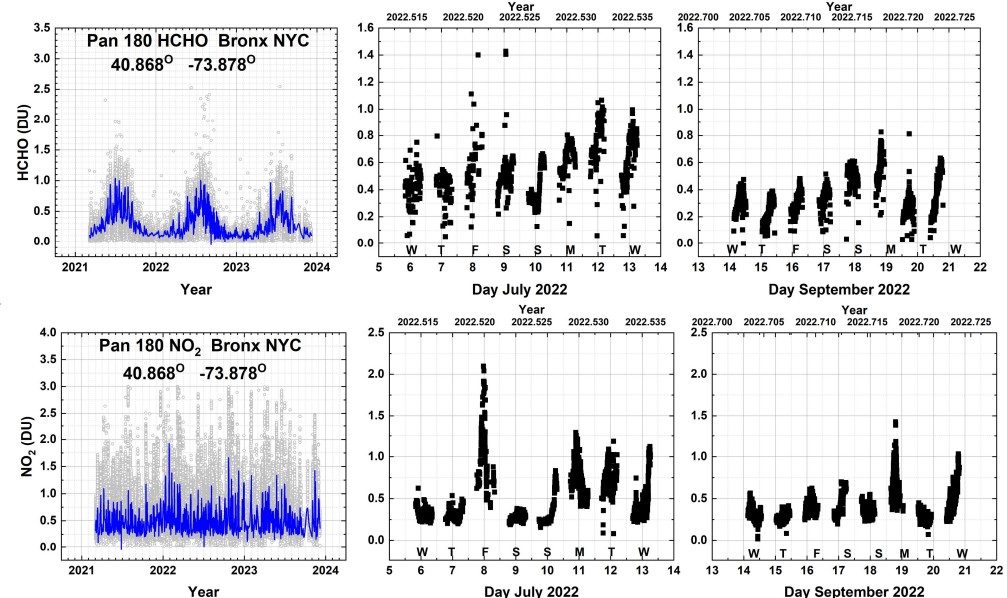

Fig. 01 Seasonal and daily behavior of HCHO and NO$_2$ from Pan 180 located in the Bronx, NYC at 40.868$^O$N, -73.878$^O$W. The blue lines are a Lowess(0.033) fit to the data (light grey), which is approximately a 1-month local least-squares average. The Local principal investigator for Pan 180 is Dr. Luke Valin.


**Figure 01**

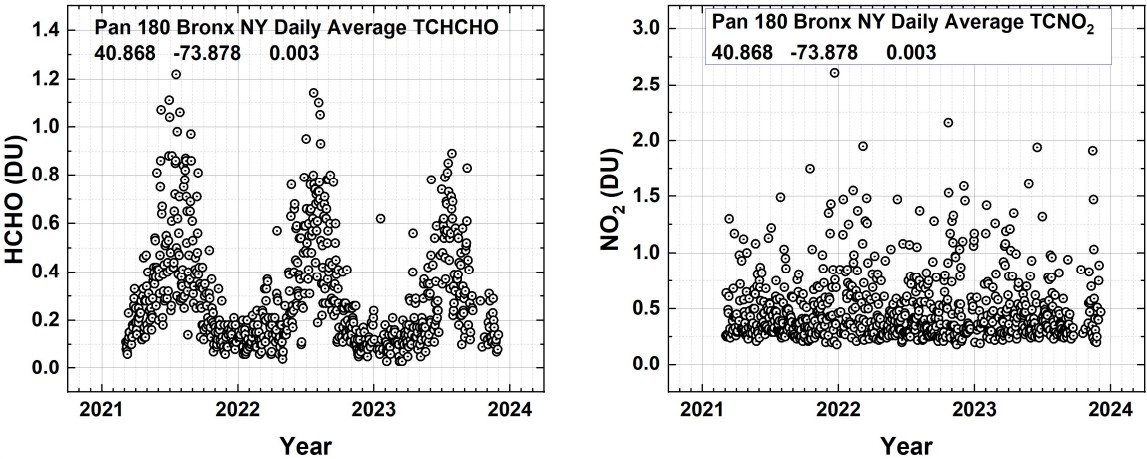

Fig. 02 The daily average seasonal variation of TCHCHO and TCNO2 in DU over Fordham University in Bronx, New York City from Pandora 180 at 40.868$^O$ latitude, -73.878$^O$ longitude, and 0.003 km altitude. Each point is a daily average of the data in Fig.1. Local principal investigator: Dr. Luke Valin.

**Figure 02**

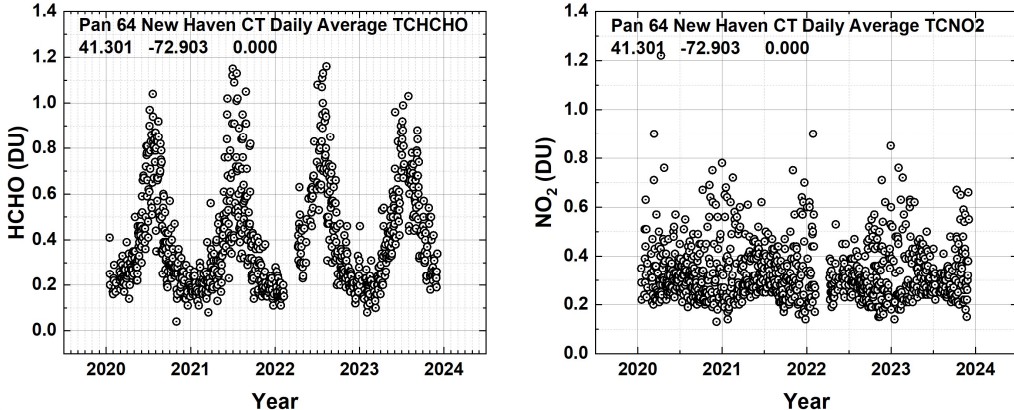

Fig. 03 The seasonal variation of TCHCHO and TCNO2 over New Haven Connecticut from Pandora 64 at 41.301$^O$N latitude and -72.903$^O$W longitude. Each point is a daily average. Local principal investigator: Dr. Nader Abuhassan



**Figure 03**


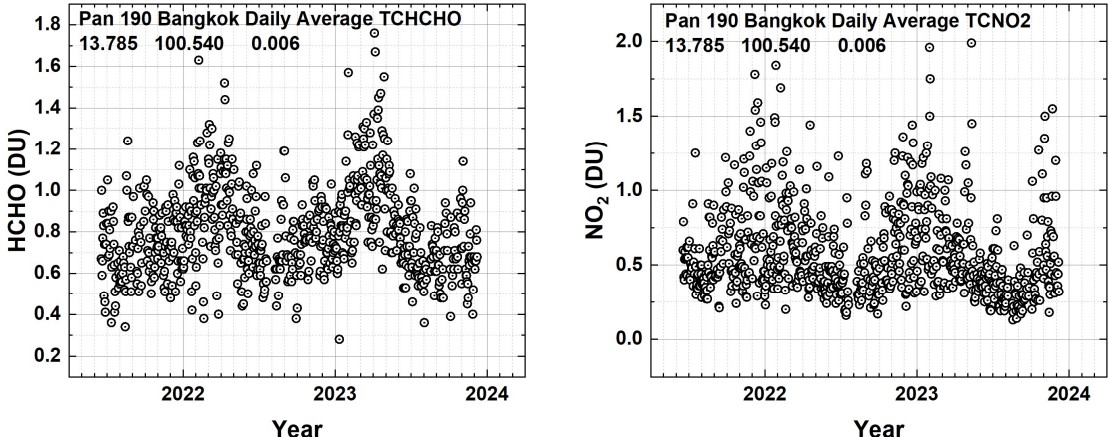

Fig. 04 The seasonal variation of TCHCHO and TCNO2 over equatorial Bangkok Indonesia at 13.785$^O$N and 100.540$^O$E. The local principal investigator is Surassawadee Phoompanit.


# Figure 04


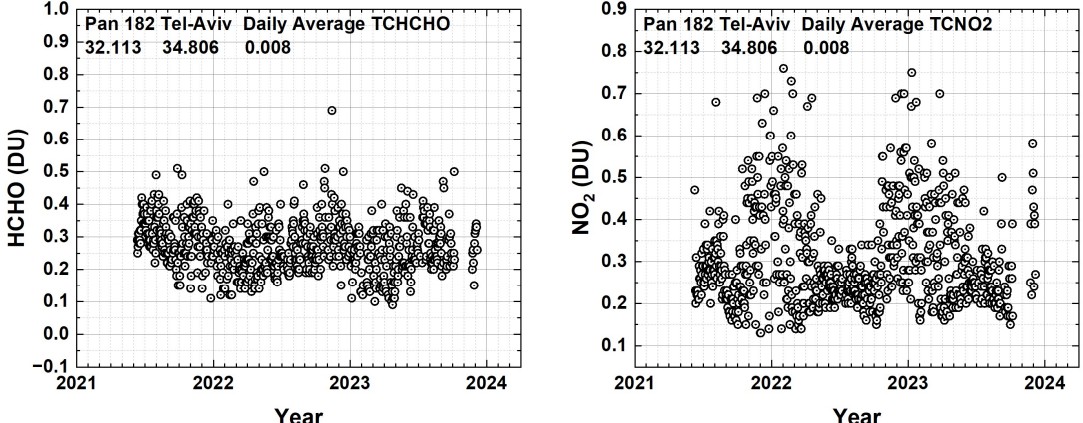

Fig. 05 Seasonal variation in daily average TCHCHO and TCNO2 in Tel Aviv Israel from Pandora 182 located at 32.113$^{O}$N, 34.085$^{O}$E at a height of 8 meters. The local principal investigator for Pan 182 is Dr. Michal Rozenhaimer.


**Figure 05**


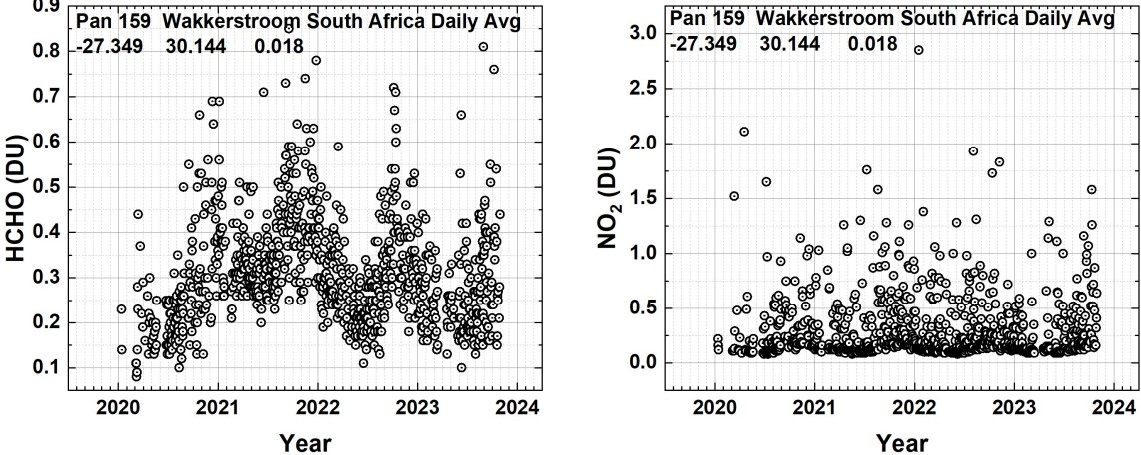

Fig. 06 Seasonal variation in daily average HCHO and $NO_2$ in Wakkerstroom South Africa from Pandora 159 located at $-27.359^O$S and $30.144^O$E at a height of 18 m. Local principal investigator: B. Scholes


## Figure 06


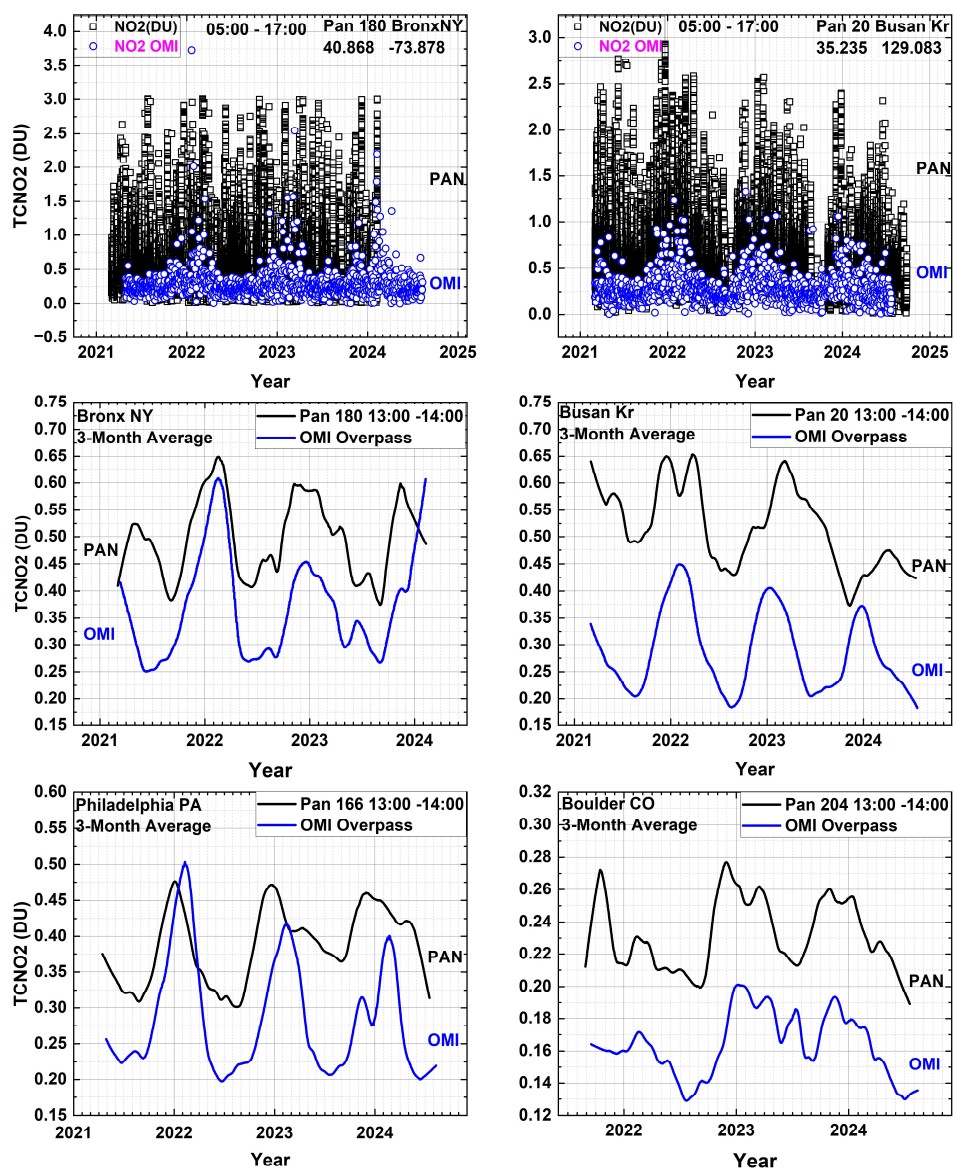

Fig. 07 Upper 2 Panels: Comparison of OMI (approximately 13:30) and Pandora (07:00 – 17:00) total column NO$_2$ time series in Bronx NY (40.868$^O$N, -73.878$^O$W) and Busan Korea (35.235$^O$N, 129.083$^O$E). Lower 4 Panels: Pandora data for Bronx, Busan, Philadelphia (39.992$^O$N, -75.081$^O$W) and Boulder (40.0375$^O$N, -105.242$^O$W) are averaged between 13:00 – 14:00 hours. Both OMI (blue) and Pandora (black) then have a Lowess(3-month) low-pass filter applied. Local principal investigator for Pan20 is Jae Hwan Kim, for Pan 180 and Pan 166 is Dr. Luke Valin, and for Pan 204 Dr. Nader Abuhassan.

**Figure 07**

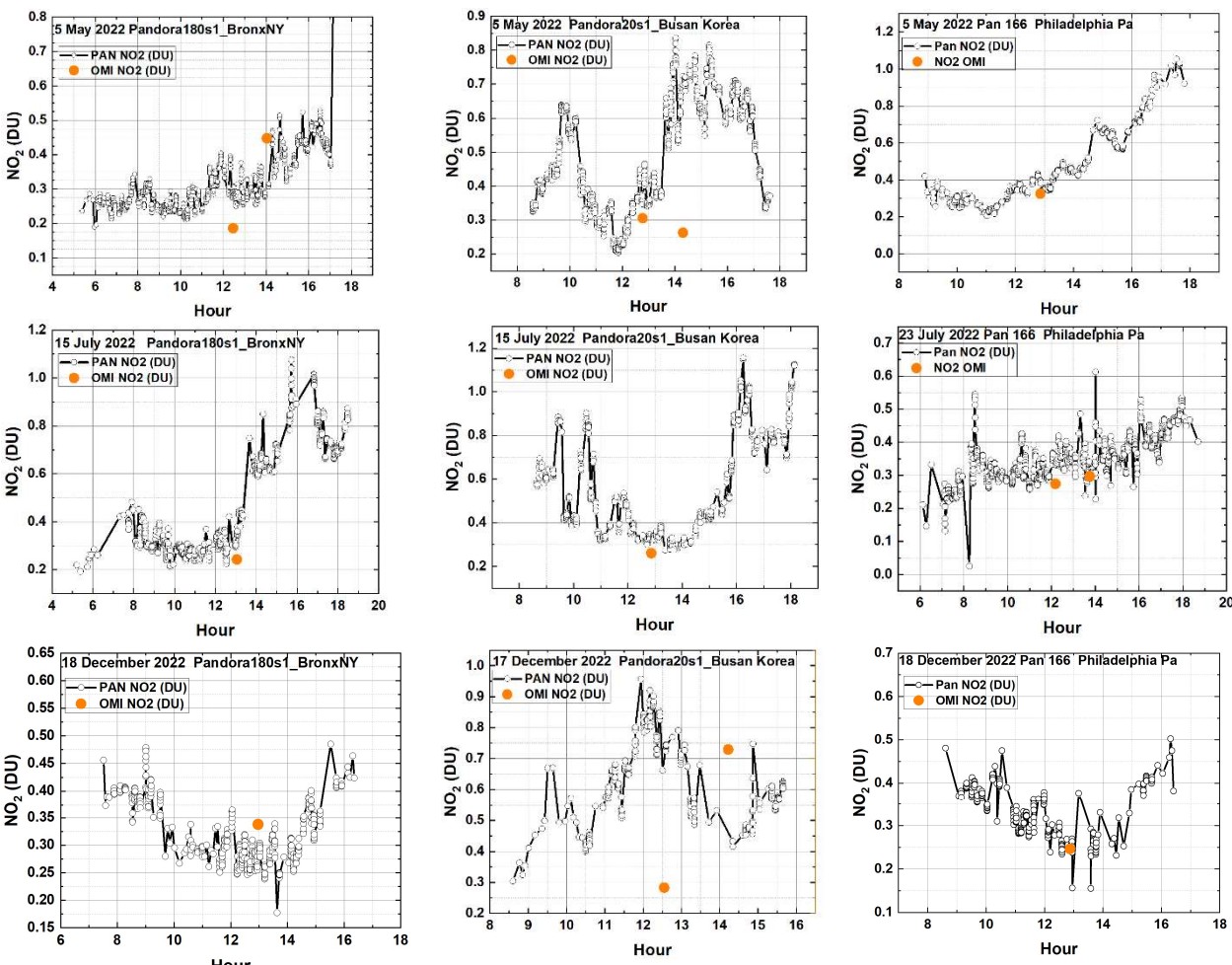

Fig. 08 A comparison between Pandora and OMI (Orange circle) total column NO₂ for 3 locations (Bronx, New York, Busan Korea, Philadelphia, Pennsylvania. The Local principal investigator for Pan 180 and Pan 166 is Dr. Lukas Valin and for Pan 20 is Dr. Jae Hwan Kim.

**Figure 08**

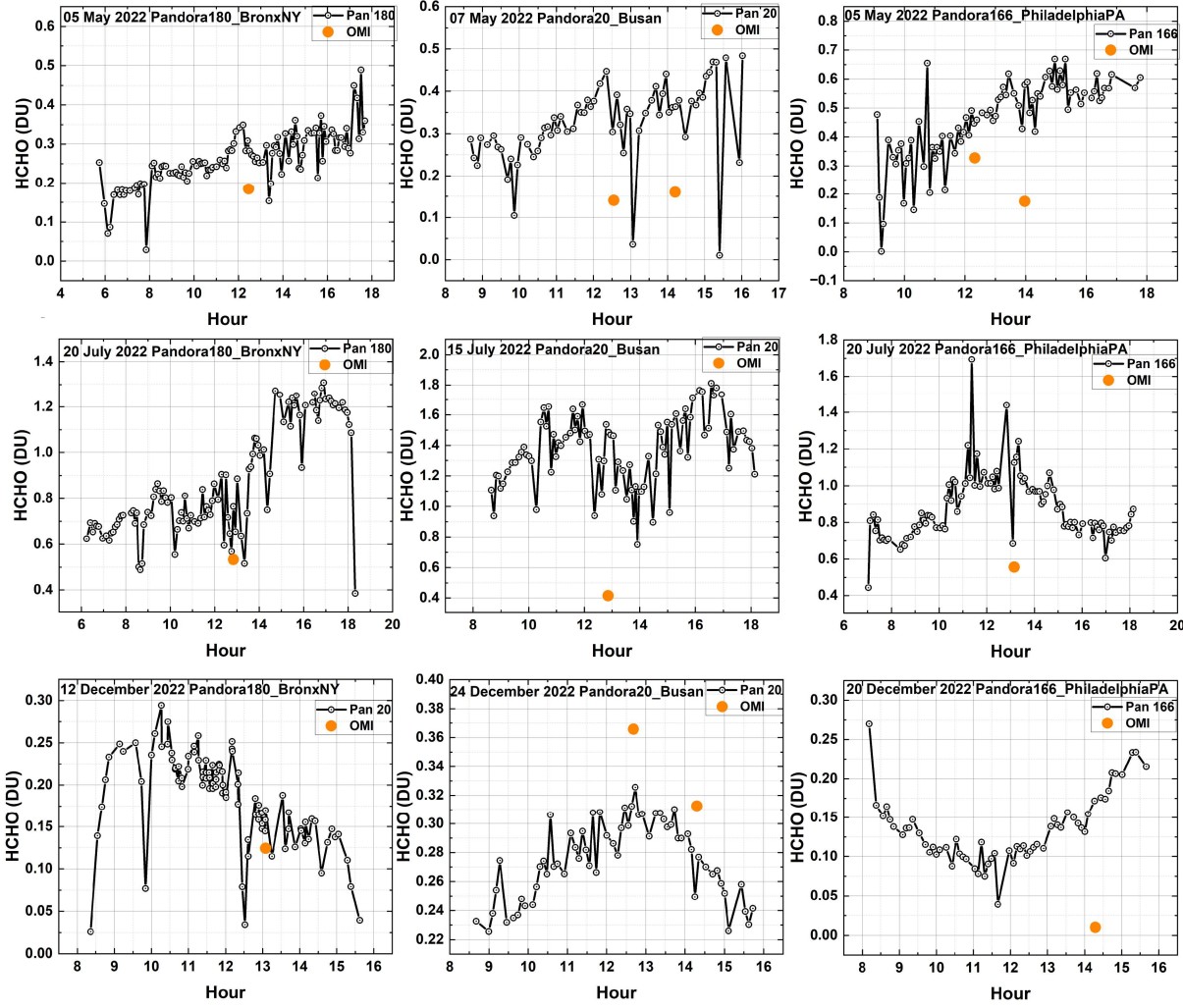

Fig. 09 A comparison between Pandora and OMI (orange circle) total column HCHO. The Local principal investigator for Pan 180 and Pan 166 is Dr. Luke Valin and for Pan 20 is Dr. Jae Hwan Kim.

**Figure 09**

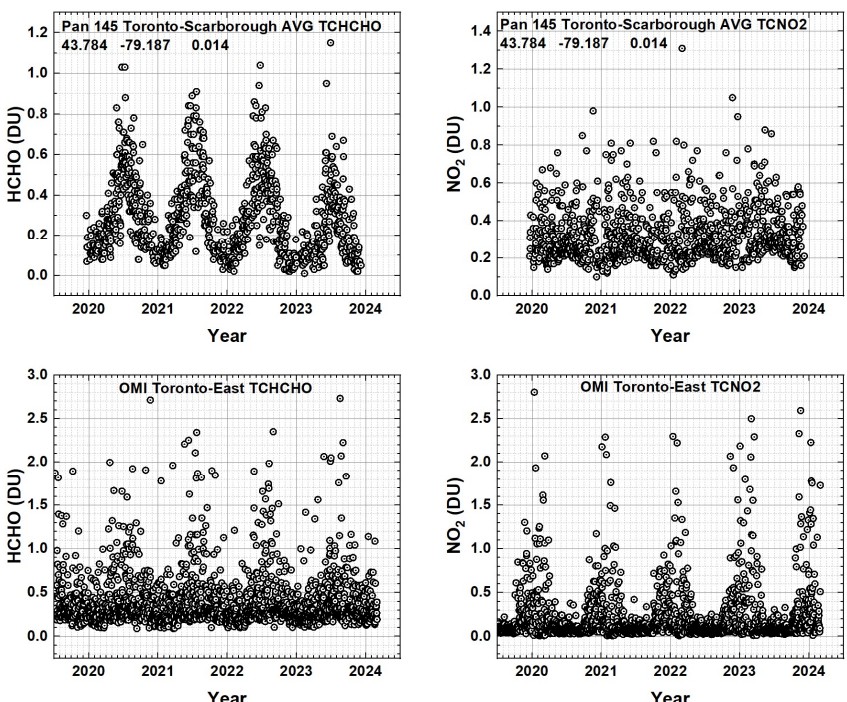

Fig. 10 A comparison of Pandora TCHCHO and TCNO2 daily average total column amounts for Toronto-Scarborough University of Toronto and OMI data for Toronto East (43.740$^O$N, -79.270$^O$W at approximately 13:20±0:20 Local Sun Time, GMT + Longitude/15). The local principal investigator for Pan 145 is Dr. Vitali Fioletov.

**Figure 10**

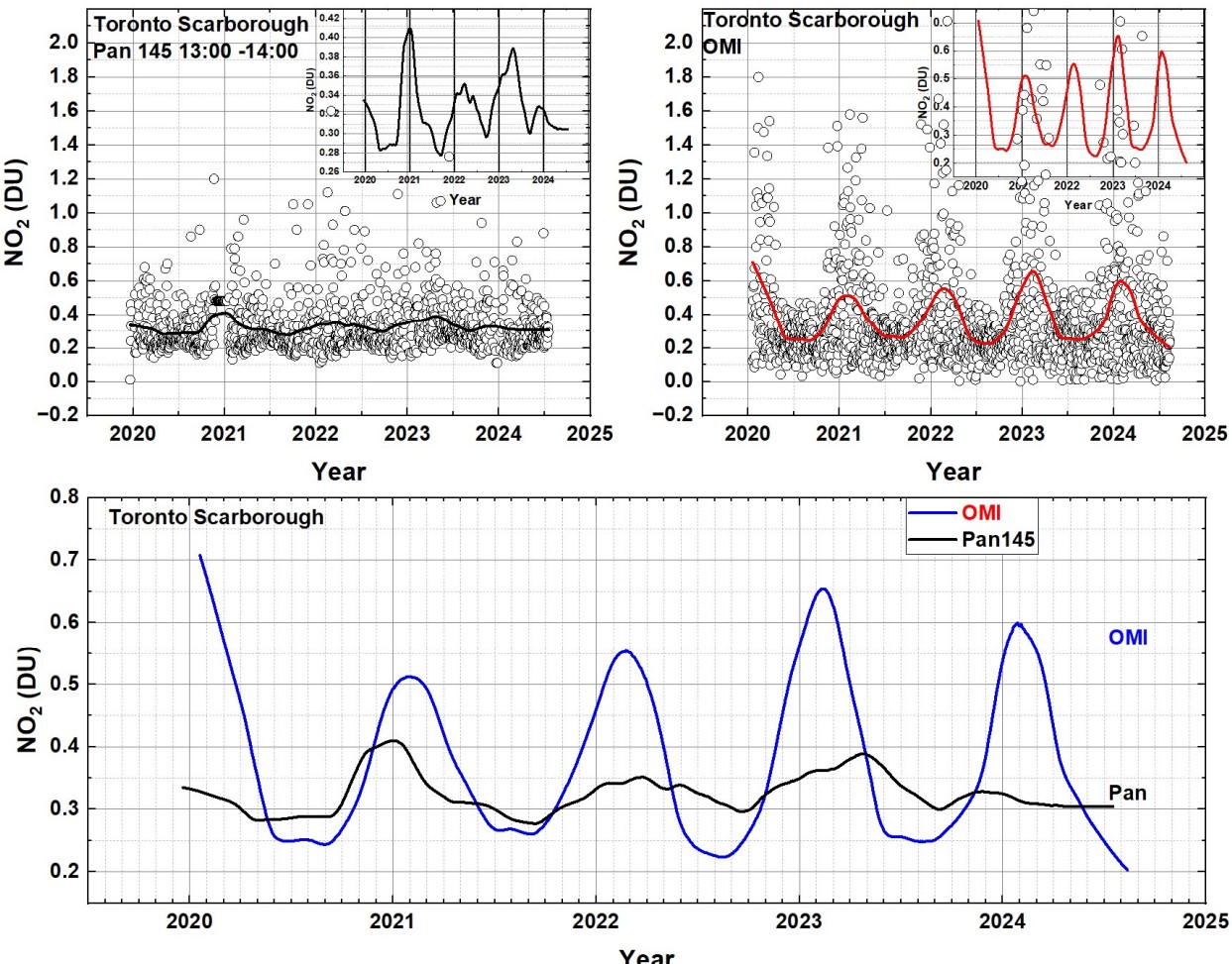

Fig. 11 TCNO2 annual cycle for Toronto Scarborough from Pan 145 average between 13:00 and 14:00 and OMI. The smooth curves are Lowess(6 Months).

**Figure 11**

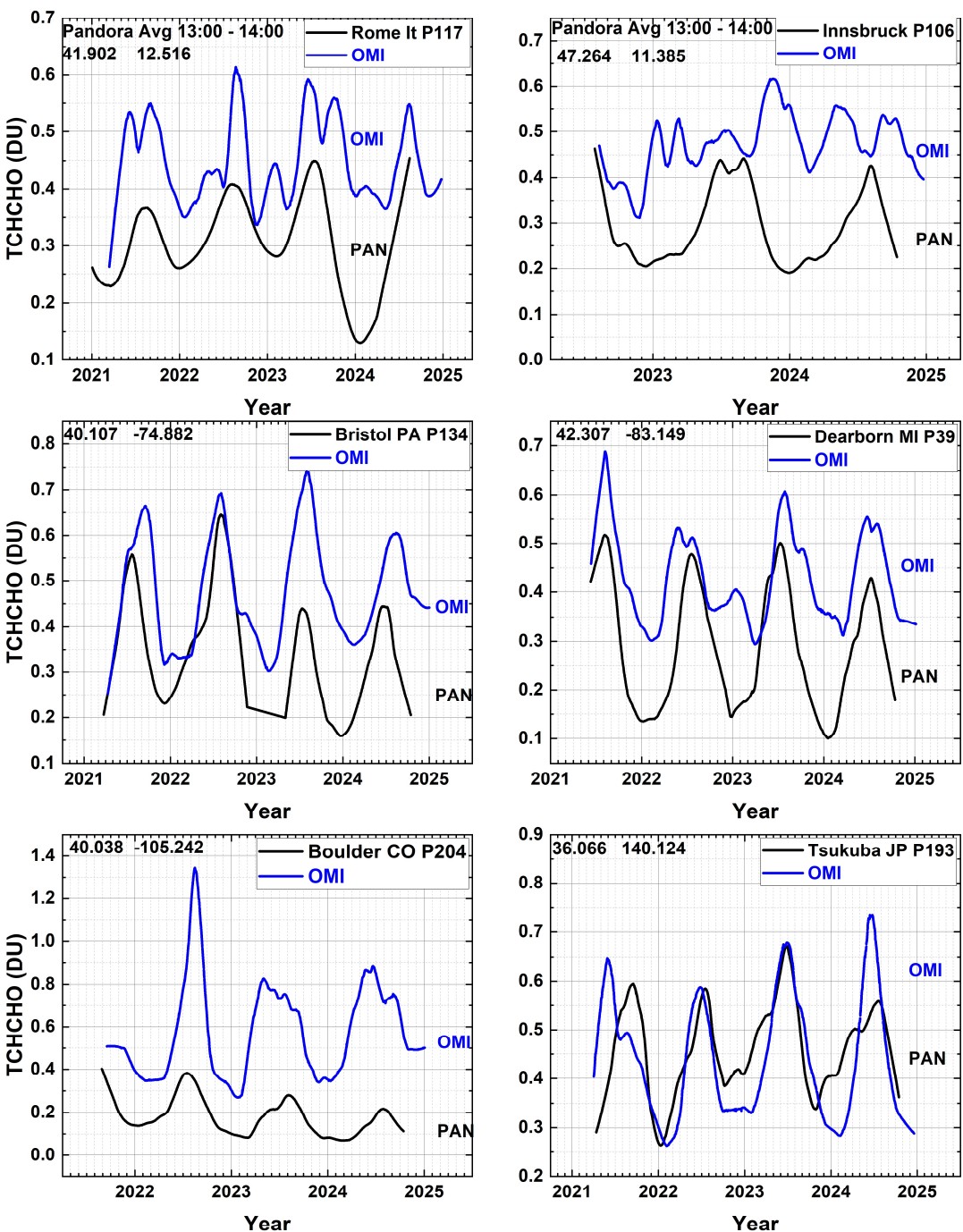

Fig. 12 A comparison between low-pass filtered, Lowess(3 months), OMI and Pandora at six sites with varying degrees of agreement with TCHCHO(Pan) < TCHCHO(OMI). The Local Principal Investigators are P106 Dr. Stefano Casadio, Dr. Kei Shiomi P193, Dr. Alexander Cede P204, Dr. Lukas Valin P39; P134, and Dr. Martin Tiefengraber P106. Latitudes and longitudes are in each upper left corner.

**Figure 12**

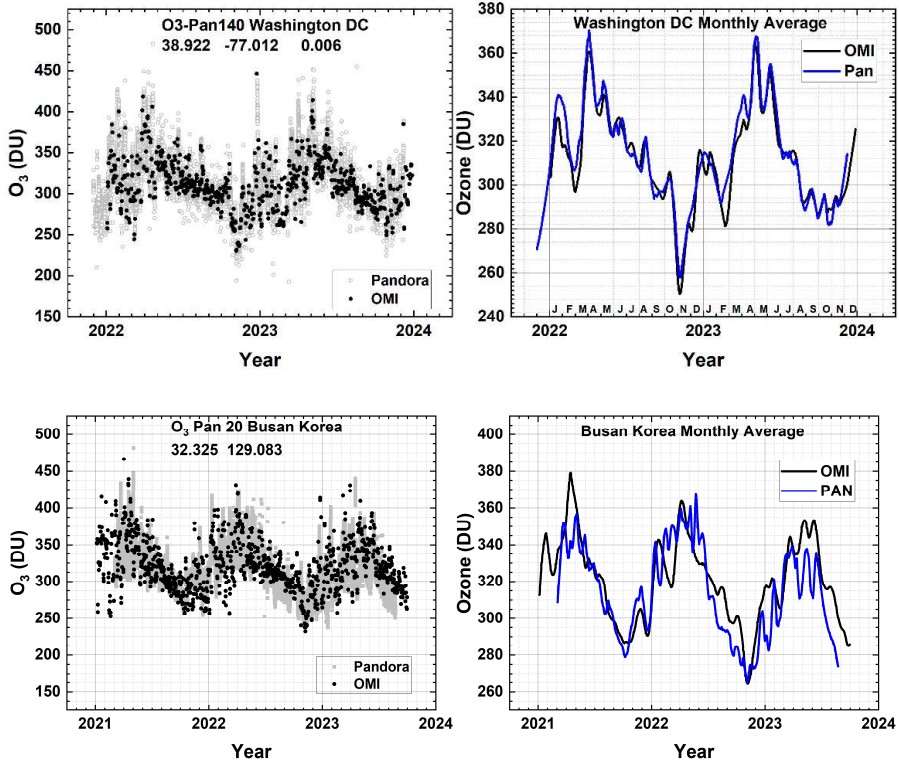

Fig. 13 A comparison of OMI Total Column Ozone values with those obtained from Pandora 140 over the Washington DC site at 38.922$^O$N and -77.012$^O$W and with those obtained from Pandora 20 over the Busan, Korea site at 32.325$^O$N and 129.083$^O$E. The smooth curves (right panel) are Lowess(6-month) fits to data in the left panel. The local principal investigator for Pan 140 is Dr. Jim Szykman and for Pan20 is Jae Hwan Kim.

**Figure 13**

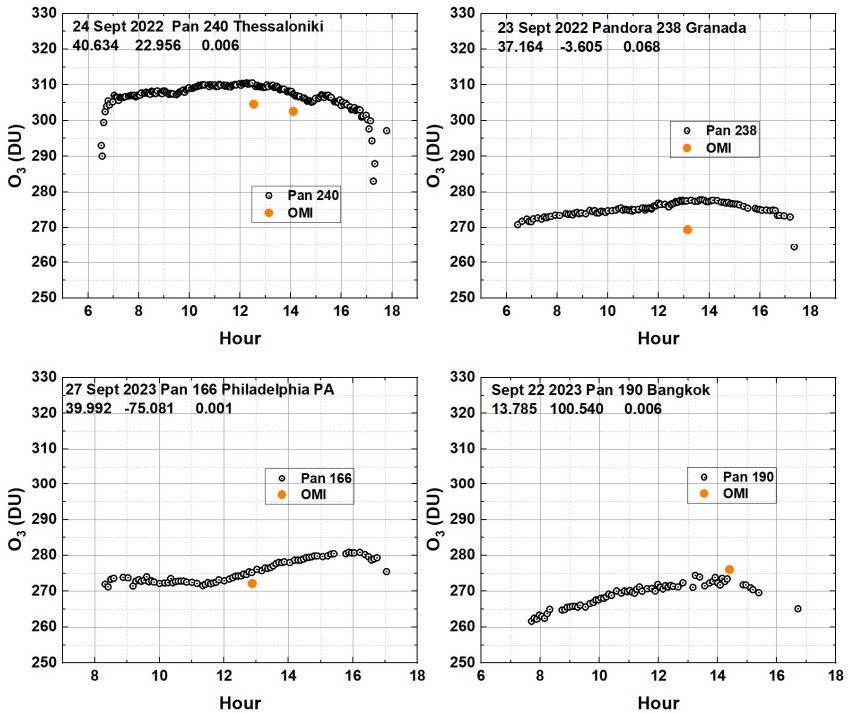

Fig. 14 A comparison of Pandora and OMI retrievals of total column $O_3$ at the time of the OMI satellite overpass. Local Principal Investigators: Pan 240 Alexander Cede, Pan 238 Inmaculada Foyo Moreno, Pan 166 Lukas Valin, and Pan 190 Surassawadee Phoompan.

**Figure 14**

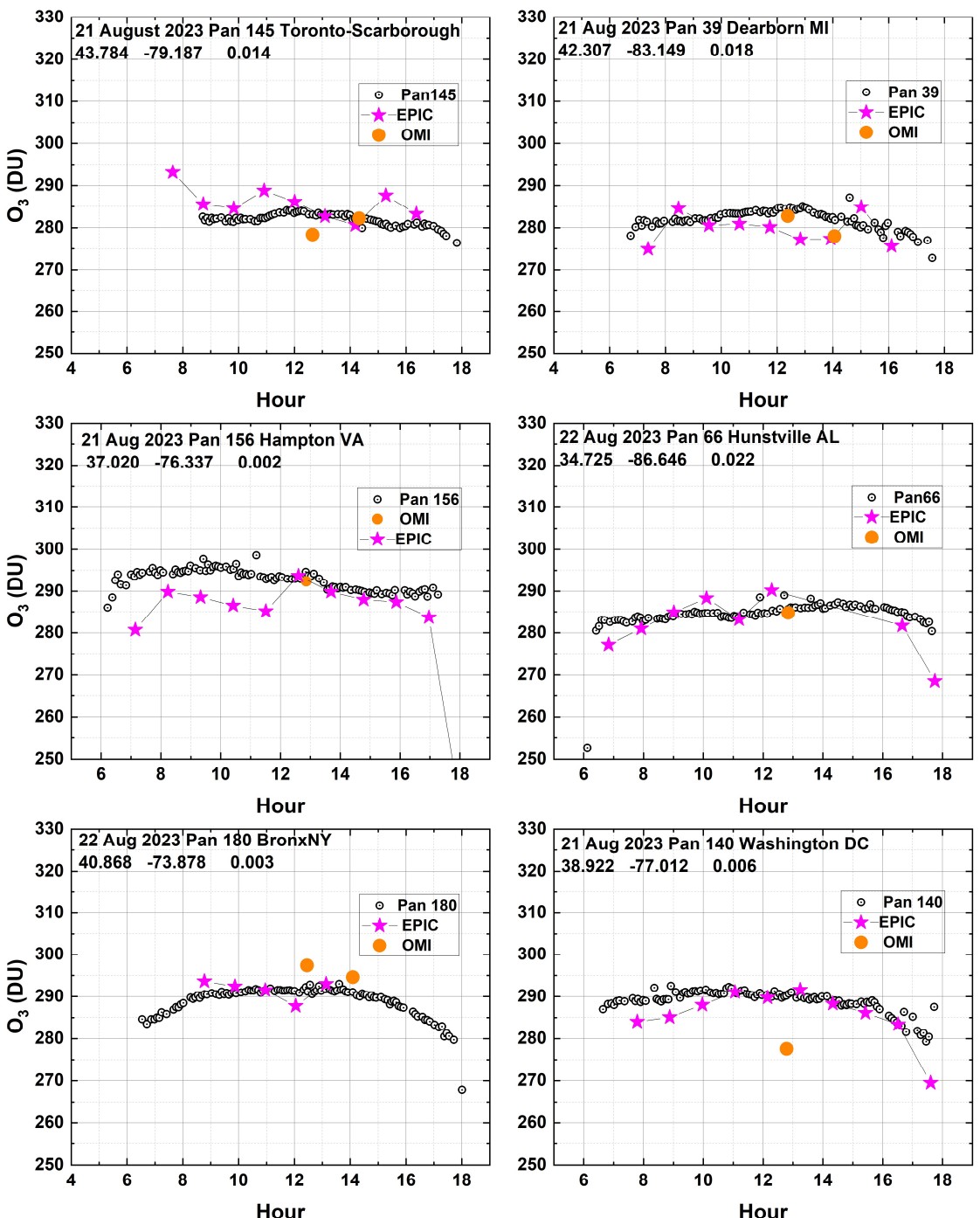

Fig. 15 A comparison of Pandora (Open Circles), EPIC (magenta stars), and OMI (orange circles) retrievals of total column O$_3$ at the times of the satellite overpasses. Latitude, longitude, and altitude (km) are in the upper left corner. Local Principal Investigators: Pan 145 Vitali Fioletov, Pan 66  Lukas Valin, Pan 39 Lukas Valin, Pan 156 Alexander Cede, Pan 66 Nader Abuhassan, Pan180 Lukas Valin, and Pan 140 Jim Szykman

**Figure 14**

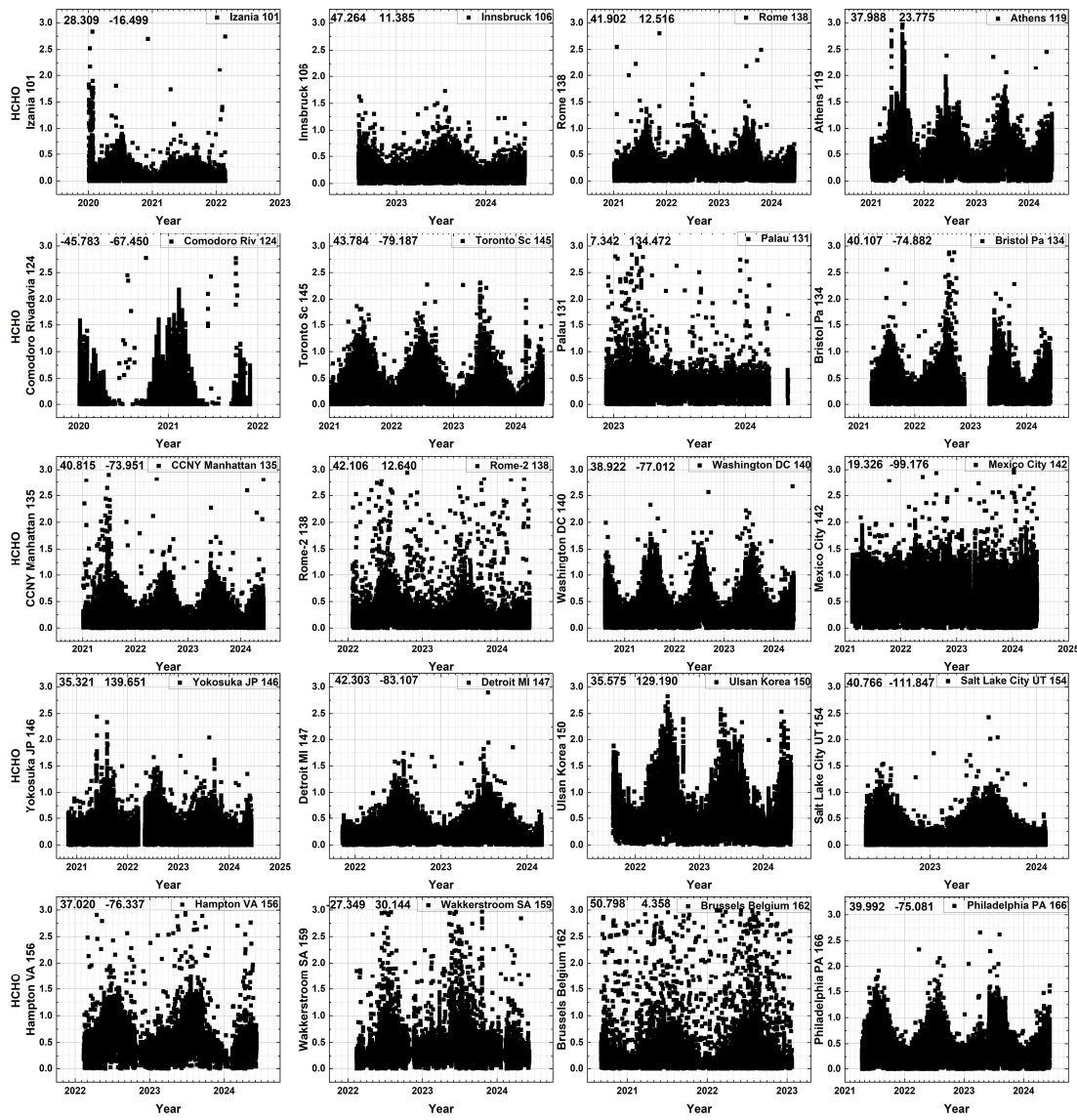

Fig. A1 The seasonal cycle of TCHCHO in DU from 20 randomly selected Pandora TCHCHO time series. The numbers in the upper left corner are the latitude and longitude in degrees and the Pandora instrument number in the right corner.

**Figure A1**

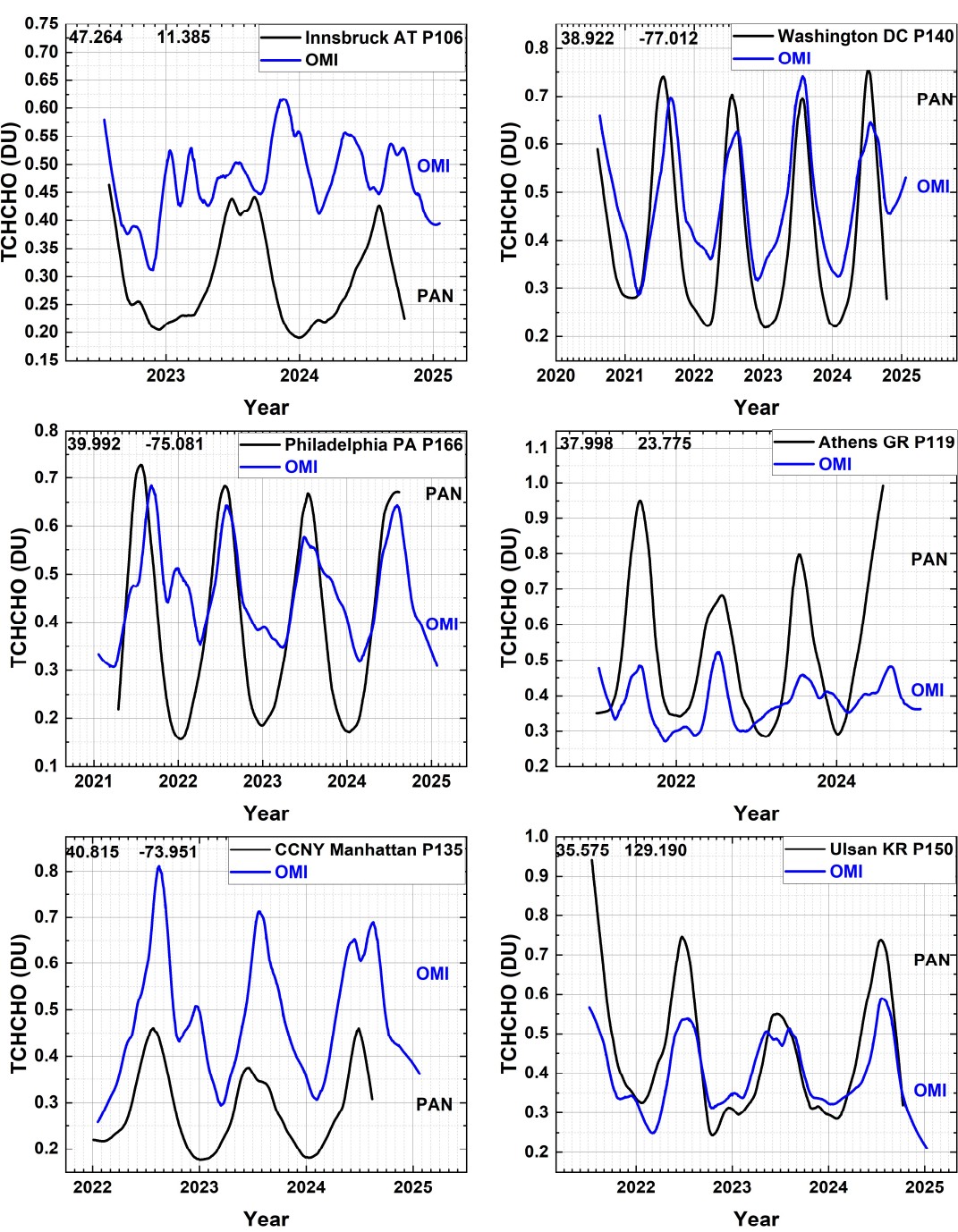

Figure A2 Six cases from Fig. A1 that have significant seasonal variation in TCHCHO. The numbers in the upper left corner are the latitude and longitude in degrees and the Pandora instrument number in the right corner. Principal Investigators are: P106 Dr. Martin Tiefengraber, P140 Dr. Jim Szykman, P166 Dr. Lucas Valin, P119 Dr. Stelios Kazadsi, P135, Dr. Maria Tzortziou, and P150 Dr. Chang Keun Song.

**Figure A2**