# Peer review of "Seasonal Variation of Total Column Formaldehyde, Nitrogen Dioxide, and Ozone Over Various Pandora Spectrometer Sites with a Comparison of OMI and Diurnally Varying DSCOVR-EPIC Satellite Data Jay Herman1,2 and Jianping Mao2,3 1GESTAR II University of Maryland Baltimor"

_EGUsphere, 2024_

## Author Comment (AC1)

Summary: This manuscript compares OMI columns to Pandora columns for formaldehyde, nitrogen dioxide, and ozone. Section 2.0 references 145 Pandoras, but figures show data fewer. The authors highlight that TCNO2 has a diurnal profile that requires time-pairing when comparing Pandora to satellite. They also highlight that OMI THCHO lacks seasonality that is expected and seen in Pandora.

Response overview:

This manuscript is not ready for publication. The goal appears to be evaluation the OMI dataset using Pandora and EPIC. It needs to be better organized, with more methodological details, and improved quantification. There are a combination of inconsistent methods (Loess vs monthly avg), statements that seem to imply methods that are neither discussed nor have results reported. One of the current conclusions seem like they would be removed if the methods had been more appropriate. With additional methods and quantification, this will be a nice contribution to assessing the satellite assets currently monitoring our atmosphere.

- The introduction does not currently describe the motivation for the study and needs  reorganization. The introduction discusses sources of pollutants and closes with the idea that it will compare OMI and Pandora. It does not describe why a comparison of OMI and Pandora would be useful. It appears to be a validation paper, which is good but is not clear. If this is a validation paper, the paper should include descriptive statistics (and more stations).

- The manuscript needs a methods section.

  - The authors need to describe the Pandora data selection and filtering. **Filtering done by error estimate and distance from Pandora site**

  - The authors describe example file names, but do not discuss their meaning. As a Pandora user, I am aware that rnvs means direct sun NO2 and rfus means sky-scan HCHO, but the average reader may not.

**rfus     = direct sun HCHO   not sky scan      rfuh = sky scan HCHO**

  - This raises the question why are you using sky-scan measurements for HCHO? And, how were stations selected as having sufficiently high quality sky-scan HCHO for comparison? **Not using sky scan**

  - Which stations are used and where are they? Section 2.0 references 145, the acknowledgements references 63, and I did not count the number referenced in the

figures. Given that conclusion statements like "most sites" are made, it should be clear which sites were part of the analysis.

I have added **Table 1 List of Pandora locations used in this study in order of appearance**

   - How OMI and Pandora were paired. Are OMI pixels within a certain distance used? Or only when the Pandora site is within the pixel geometry defined by the corners?

**OMI pixels within 50 km filtered by my Fortran program**

  - The OMI product is insufficiently described in the document. There are no version numbers or citations. The authors have provided the URL of where to get OMI, but nothing about the data product or when they acquired the data. Websites change and the Aura website will likely be lost after Aura is decomissioned. **The AVDC website is not dependent on the AURA mission.** More details about the data product need to be included in the publication for posterity. According to https://ozoneaq.gsfc.nasa.gov/products/ozone/, "Overpass (OVP) products are a weighted average of data within a defined range to a set of ground station locations." What the distance of the defined range? There is a 50km version that is very clear about its distance, but the README for the standard OVP products is not clear.

  - If you're going to use a Loess fit, it would be good to introduce it somewhere. The Loess (not Lowess right?). **No Lowess, the Loess routine is different**. **I gave a reference.** fit seems unnecessary given that all your other plots use simple running means. **My mistake in writing. Lowess is similar in purpose to a running average but reduces the effect of outliers.** Perhaps you could explain why it is appropriate for Pan 180 NO2 in Figure 1, but not Figure 7. **All smooth curves are Lowess**

**Figure 7 caption now reads. "Both OMI (black) and Pandora (red) then have a Lowess(3-month) running average applied."**

**Figure 11 now has "The smooth curve is Lowess(6 Months)."**

- Figures are often scatter plots where the markers are so dense that often only a cluster is visible. The distribution of values is not decipherable. I recommend creating some sort of synthesis plots. **I am not sure what a "synthesis" plot might be. Other than adding a Lowess fit, I do not see the value of a different form.**

- The longer time-series would benefit from some sort of statistical analysis that quantified "agreement" and "disagreement" in the abstract. Right now, there is little more than visual analysis of datasets that were processed by others and downloaded.

**I assume that you are referring to the ozone timeseries in Figs 12 and 13**

- Similarly, it would be nice to quantify seasonality. XX% higher in JJA than DJF or similar. The conclusions starts with a paragraph about seasonality, but right now the manuscript simply says it is seen in one dataset and not the other.

**That is correct. The amount of seasonal variation for TCHCHO varies depending on the site.**

**For most midlatitude sites, the seasonal variation is significant and occurs during summer.   This has been added to the Summary**

- The idea that Pandora would agree best if paired in time seems like an obvious conclusion. Figure 7 and analysis could be simplified by highlighting (or citing) the diurnal variation in the methods sections as the reason for time-pairing.

- The conclusion that OMI "underestimates" the degree of atmospheric pollution does not seem novel or quite accurate. OMI only "underestimates" pollution if we assume that overpass (13:30LST) is representative of the whole day. We know that vehicular emissions clearly peak at rush hour, so we would expect columns not to peak at Aura overpass (13:30LST). There is much evidence of this understanding in the literature. For example Anenberg et al. (doi:10.1016/S2542-5196(21)00255-2) use a series of ratios to translate overpass-time data to daily averages (see Figure S1 and discussion).  The submitted manuscript should cite existing works highlighting the coincidence of the local minimum as a need for temporal co-sampling rather than highlighting this as a finding. **Added Lamsal, L. & Duncan, Bryan & Yoshida, Yasuko & Krotkov, Nickolay & Pickering, Kenneth & Streets, David & Lu, Zifeng. (2015). U.S. NO2 trends (2005–2013): EPA Air Quality System (AQS) data versus improved observations from the Ozone Monitoring Instrument (OMI). Atmospheric Environment. 110. 10.1016/j.atmosenv.2015.03.055**.

- The PGN website requests that "The PGN is a bilateral project supported with funding from NASA and ESA." be added to the acknowledgements. https://www.pandonia-global-network.org/home/documents/pgn-data-use-guidelines/   **Done**

Line-by-line notes:

 - 35: the abstract discusses seasonal dependences, but isn't clear one what would be "big" or "little".

 - 36-39: the abstract and conclusions assert that OMI is not observing near the surface, but the authors only show that it fails to capture seasonality. Could the failure have to do with reference sector correction? Or some other failure? Could you explain why you specifically think it is a failure to sense the lowest levels.

**Since both Pandora and surface measurements (Wang et al., 2022) see the seasonal dependence of HCHO it is likely that OMI is not seeing into the boundary layer or is averaging over non-vegetated areas. The conclusion has been modified.**

. For most sites, OMI does not observe the strong seasonal variation of TCHCHO that is clearly seen in the Pandora data and in surface measurements (Wang et al., 2022).  The lack of OMI seasonal variation in TCHCHO at most sites suggests that OMI may not be seeing the lowest layers of the HCHO variation or may be averaging over non-vegetated areas. The amount of seasonal variation for TCHCHO varies depending on the site. For most midlatitude sites, the seasonal variation is significant and occurs during summer.

 - 39-41: It seems obvious that excluding rush-hour from the comparison with 13:30 would be good. Why is this a noteworthy finding? (see discussion above)

**The abstract now has a sentence:** "Even when Pandora data is averaged between 13:00 and 14:00 hours local time OMI underestimates TCNO2." **One can see this in figure 7**

 - 43-44: Agreement and disagreement should be put into some sort of context. Is there a pattern (under-estimating high values, over-estimating at low latitude) or is it random? What does agreement mean (bias within X? correlation above Y?)?

 - 45: Does EPIC provide any particular meaningful result?  **EPIC ozone retrievals are consistent with Pandora and suggests that the calibrations are consistent**.

- 54-55: Is the point that most methane that later forms HCHO comes from these sources? Or is this arguing that the majority of HCHO comes from this specific pathway (more so than isoprene + other methane sources)? **The results from this data paper do not address this problem**

- 70: Are these citations only for the first half of the sentence? If so, what are the citations for the rest and their ranking? **The citations apply to the entire sentence and have now been moved to the end**

- 79-104: This discussion does not mention surface monitors that sample in situ air or airborne in-situ measurements. The apparent focus is column integrals, but the sentences that introduces it simply says "typically measured by." Given that surface monitors and in-situ air sampling are more common, I think this needs clarification. **The sentence now reads "TCHCHO, TCNO2 and TCO in the atmosphere are typically measured by satellite and ground-based instruments."**

- 83: Given the timing of this submission, it is worth noting the TEMPO and GEMS satellites if this is a list. If this is really the methods section, then I don't think you use several of the data sources in this list. **This paper was written and submitted before TEMPO data were available. GEMS has few underlying Pandora sites. A paper discussing geostationary time dependent data is being written.**

- 117: seems weird to note that the website is Austrian. **Now reads "Austrian project website"**

- 119-122: Rather than providing file names, perhaps it would be better to discuss the meaning of the codes. For example, my read of the names is that you're using direct sun for the NO2 and skyscan for the HCHO. However, you do not discuss that. You also make no mention of data filtering of any kind. **I only used direct-sun data as mentioned earlier. The file names are all direct sun. The OMI overpass filtering is for data within 50 km of the Pandora location. All of the comparisons for OMI data are mid-day so that solar zenith angle filtering is not needed. Pandora data for single-day comparisons are selected for those days where it is likely that Pandora is observing clear sky.**

- 126-127: Did you use their other measurements? **I only used the direct-sun Pandora data as mentioned,  rfus5p1-8, rnvs3p1-8, and rout2p1-8**

- 129-136: Using a single week a representative of day-of-week distributions is not a good idea. Friday July 8 might have been an outlier with winds blowing from a specific

source that was active but downwind on Thursday. Why is this specific week a good case study? **I was not doing a case study, just showing two examples. Other weeks are somewhat different. I picked a week in July and a week in September.**

- 133: "summer seasonal dependence" should probably be "summer peak"? **No**

- 197: To me, it looks like NO2 peaks in the DJF period in the lower left and lower right plots for OMI (green line). Perhaps the seasonality in OMI is larger than the seasonality in Pandora. **The TCNO2 peaks are in DJF and track each other for PAN and OMI with Pandora values > OMI values. The new Figure 7 shows this more clearly.**

- 206-207: "not statistically different" -- this stands out to me because I do not see any statistics anywhere. Nor do I see a discussion of how differences will be tested for significance. t-test? Welch's? Mann-Whitney? To see a statement like this, I would expect to see some data characterizations (mean+-std) for both datasets and/or the differences. **You are right. That sentence has been removed as being meaningless**

- 211-212: Can you at least state which Pandora sites you looked at? **See Table 1**

- 217: How did you measure the "cases of agreement"? For example, did you consider the uncertainty in either measurement? **I left out the criteria, now added within 10%.**

- 226-233: Is this the only site where the OVP file is not aligned with the Pandora site? If so, why is it a good site to show? **OMI Toronto East is Latitude:   43.740 deg. Longitude:  -79.270 deg. The Pandora site is at 43.740$^O$N, -79.270$^O$W They are exactly aligned**

- 234: Why are 10a and 10b not 10 and 11? **They now are. Figures have been modified and renumbered**

- 234: Figure 10a -- what is the AVG for Pandora? Is this 13:50 to 14:50? Or some other window? **13:20 +/- 0:20 Local sun time  (GMT + Longitude/15) Now in Figure caption**

- 235-241: The discussion of highway vs heating seems speculative. **The highway has nothing to do with heating. NO2 is produced continuously from traffic. The heating is seasonal but is mostly electrical heat and not from burning gas. Therefore, no seasonal local NO2 emission from heating.**

- 250: There is currently no discussion of Figure 11 or 12. **The discussion is above the figures (now 12 and 13) and is just to show that the calibration of the Pandoras is consistent with OMI.**

- 253: "good agreement ... at most sites" - is this comment based on the 4 sites shown? or was more analysis done? It is good that here there is a statement about what the difference is, but are we really talking about just one day? **More sites have been looked at and a separate paper has been written on ozone comparison for a special issue on EPIC. This is just a sample that was not selected for good or bad agreement.**

- 267: "for most sites" should be quantified. Of the N sites, M do not... I say this because the paper uses a small set of Pandora as representative. The conclusion could be interpreted as "most" Pandora sites. This may be true, but the paper does not show that. Instead it relies on a few (3?) case study sites.

**The number of TCO comparisons has been expanded to 10 sites (Figs. 14 -15), all showing fairly good agreement. While this does not prove that all Pandora sites are good (6 are known to have stray light problems) it is representative of most of the sites in the Pandonia website that produce TCO data.**

**Citation**: https://doi.org/10.5194/egusphere-2024-1216-RC2

---

## Author Comment (AC2)

General Comments:

The paper by Herman and Mao is a study comparing Total Column HCHO, NO2, and O3 from Pandora Spectrometers to OMI and DSCOVER-EPIC. They included multiple pandora stations located at various locations around the globe and during different seasons. They found that agreement is overall good, however OMI does not alawys capture the seasonal variation as seen in the pandoras and may not be sensitive to changes in surface concentrations. DSCOVER-EPIC agrees quite well with the diurnal pandora data. This is a much needed comparison study as there are few publications on the validaty of pandora spectrometers which are to be used in future satellite validation plans. The manuscript requires some minor changes as well as some additional discussion/figures before publication.

Specific Comments:

79-- Introduction mentions airborne data but does not include any in results. I would be interested to see the comparisons. Otherwise remove from introduction.

**Removed**

83-- Why not use TROPOMI in this study instead of (or alongside) OMI? It is mentioned but not used.

**TropOMI Pandora overpass data are not publicly available.**

120-- The text files of Pandora do not need to be explained in such detail. However, I would like to know what data quality flags are being used to filter out bad quality data.

**I found that the Pandora file names are confusing with regard to the file contents, so I listed example names of the total column density files.**

**I used the RMS error (Column 45) to filter the data as well as flags for no spectral fitting. OMI data within 50 km of the Pandora location. Other than that, I used all of the data within the time limits specified in each figure.**

Fig 01:

-- Why not show a continuous time series for the July and September weeks?

**Figure 1 has been redone**

-- Change y axis for the NO2 day comparisons to be equal.   **Done**

-- Discuss why the Lowess line is important.

**The lowess line Lowess(f), f = 0.03, is similar to a 30-day running average except that outliers are weighted less just as in a conventional linear least squares fit. The parameter f is the fraction of the data included in the local least squares.**

**The text now reads: "**For TCNO2, there is only a weak seasonal pattern as shown in the Lowess(0.03) fit to the data (Cleveland, 1979; Clevland and Devlin, 1988) with small maxima in January-February, since the sources of $NO_2$ are largely from the nearly constant flow of cars and trucks. The parameter 0.03 is the fraction of the data included in the local least squares estimate.**"** **Green color is new text**.

Line 138-- No need to list the file name just state which station is being discussed.

**OK  Removed**

Fig 02:

-- What is the "0.003" listed on the figure? **0.003 = altitude in kilometers. The altitudes are now in Table 1 in meters**

-- Fig 2 and Fig 1 both state that there is a seasonal dependence of HCHO but not NO2. Fig 1 is not necessary unless the daily panels are further discussed.

**Figure 2 reduces the noise by using daily averages**

Line 145-154-- What is the reasoning behind showing some pandora figures and not others? Why not include a a monthly average time series of all pandoras on one figure or at least group by certain locations. This would also help see the difference in magnitude of TCHCHO and TCNO2.

**I am not sure what you are asking. There are about 180 active Pandora spectrometers located all over the world. I cannot include a figure for all of them. I picked a subset that shows typical behavior including some sites that do not have a seasonal HCHO dependence. I have added Lowess smoothed figures to show the differences between Pandora and OMI**

Line 154-- Please include a table of all Pandora stations included in this study. The wording is vague about which pandora stations in CT are included in this statement.

There are several stations along the CT coastline. This will also prevent the lat/lon and PI from needing to be stated in every figure.

| | Table 1 List of 30 Pandora locations used in this study and figure of appearance | | | | |
|---|---|---|---|---|---|
| | Pandora Number | Pandora location name | Lat (deg) | Long (deg) | Alt(m) |
| 1 | Pan 180 Fig.1 | Bronx, New York USA | 40.868 | -73.878 | 31 |
| 2 | Pan 64 Fig.3 | New Haven, Connecticut USA | 41.301 | -72.903 | 4 |
| 3 | Pan 190 Fig.4 | Bangkok, Indonesia | 13.785 | 100.540 | 6 |
| 4 | Pan 182 Fig.5 | Tel Aviv, Israel | 32.113 | 34.806 | 8 |
| 5 | Pan 159 Fig. 6 | Wakkerstroom, South Africa | -27.349 | 30.144 | 18 |
| 6 | Pan 20 Fig.7 | Busan, Korea | 50.798 | 4.358 | 107 |
| 7 | Pan 145 Fig.10 | Toronto-Scarborough, Canada | 43.784 | -79.187 | 14 |
| 8 | Pan 134 Fig. 12 | Bristol, Pa, USA | 40.107 | -74.882 | 10 |
| 9 | Pan 204 Fig. 12 | Boulder, Co USA | 40.038 | -105.242 | 161 |
| 10 | Pan 106 Fig.12 | Innsbruck, Austria | 47.264 | 11.385 | 616 |
| 11 | Pan 117 Fig.12 | Rome Italy | 41.907 | 12.5158 | 75 |
| 12 | Pan 193 Fig.12 | Tsukuba, Japan | 36.066 | 140.124 | 51 |
| 13 | Pan 140 Fig.13 | Washington, DC USA | 38.922 | -77.012 | 6 |
| 14 | Pan 166 Fig.7 | Philadelphia, Pa  USA | 39.992 | -75.081 | 6 |
| 15 | Pan 238 Fig.14 | Granada | 37.164 | -3.605 | 7 |
| 16 | Pan 240 Fig. 14 | Thessaloniki, Greece | 40.6336 | 22.9561 | 60 |
| 17 | Pan 66 Fig.15 | Huntsville Alabama USA | 34.725 | -86.646 | 22 |
| 18 | Pan 156 Fig.15 | Hampton, Virginia USA | 37.020 | -76.337 | 19 |
| 19 | Pan 39 Figs.12,15 | Dearborn, Michigan USA | 42.307 | -83.149 | 18 |
| 20 | Pan 101 Fig.A1 | Izania, Spain | 28.309 | -16.499 | 24 |
| 21 | Pan 119 Fig.A1 | Athens, Greece | 37.998 | 23.775 | 130 |
| 22 | Pan 124 Fig.A1 | Comodoro Rivadavia | -45.7833 | -67.45 | 46 |
| 23 | Pan 131 Fig. A1 | Palau | 7.3420 | 134.4722 | 23 |
| 24 | Pan 135 Fig.A1 | CCNY Manhattan NY USA | 40.815 | -73.951 | 34 |
| 25 | Pan 142 Fig.A1 | Mexico City, Mexico | 19.326 | -99.176 | 2280 |
| 26 | Pan 146 Fig.A1 | Yokosuka, Japan | 35.321 | 139.651 | 5 |
| 27 | Pan 147 Fig.A1 | Detroit, Mi USA | 42.303 | -83.107 | 178 |
| 28 | Pan 150 Fig.A1 | Ulsan, Korea | 35.575 | 129.190 | 38 |
| 29 | Pan 154 Fig.A1 | Salt Lake City Ut, USA | 40.766 | -75,081 | 1455 |
| 30 | Pan 162 Fig.A1 | Brussels, Belgium | 50.798 | 4.358 | 107 |

Line 178-- Why is there a seasonal NO2 pattern if, like NYC, the pandora is near automobile sources?

**I do not know. The Pandora in Tel Aviv is located in the middle of the University not far from the coast and not in a main traffic area as is the case for NYC. There is a highway, Route 20, about 1 km from Pandora 182. The apparent peak occurs in January, which is the rainy season. Maximum power generation is during the summer, while the TCNO2 peak is in the winter. This is an observation of data for which I do not have an explanation.**

Line 191-- How were these days chosen? Is the OMI agreement dependant on the diurnal pattern of HCHO?

**The days were selected just to give a sample of OMI vs Pandora comparisons. With some days having good comparison and many days having large differences.**

Fig7:

--How were the pandoras used for the OMI comparison chosen out of a total of 147?

**The Pandoras were chosen just to give a sample of the Pandora sites. I have added a plot of 20 randomly chosen sites in the appendix**

--Where is the monthly average TCHCHO comparison figure?

**I have added 3-Month average figures for TCHCHO**

Line 203-- Reword. It isn't that OMI and Pandora TCNO2 agree more at the overpass time, it is that the overpass time is the only available data for comparison.

Changed to: **This shows that OMI and Pandora TCNO2 agree more closely when the comparison is restricted to the overpass time.**

Line 205-- What method are you using to compare OMI and Pandora. Is it a single pixel that overlaps the pandora? A given radius in km?

**A new sentence has been added**

**The original OMI data has a resolution of 13 x 24 km$^2$ at the center of the OMI side-to-side scan. The closest OMI pixel to each Pandora site is used for the time matched comparison. The largest distance is 50 km.**

Line 220-225-- If the HCHO comparison results are due to the ozone retrieval influences then what is the TCO3 at these dates? Is NO2 better because it is not impacted by ozone spectral fingerprint?

**The TCO3 and TCHCHO are retrieved by spectral fitting in an overlapping spectral range 300 to 340 nm. The retrieval method for TCO3 used in this analysis mimics the TOMS wavelength ratio algorithm that is not significantly affected by HCHO. After TCO3 is determined, spectral fitting of the residual is formed to retrieve TCHCHO. Some of the differences of OMI TCHCHO relative to Pandora may be caused by small errors in TCO3 or by reduced sensitivity in the OMI retrieval to**

**HCHO in the lowest altitudes. Retrieval of TCNO2 is also accomplished by spectral fitting in the blue wavelength range 405 – 450 nm that is not affected by ozone and has better sensitivity at the lowest altitudes because of less Rayleigh scattering than in the 300 – 340 nm range.**

Line 220-- I would be interested in seeing a scatter plot comparing 13-14:00 UTC Pandora total column with OMI for all days of these Pandora stations. That way we can see if there is a constant bias and by how much. Otherwise explain why these days were chosen out of three years of data.

**Instead of a scatter plot, I have now presented the results of a low-pass filter that shows the bias. The bias varies with the site location. The particular sample days for Spring, Summer, and Winter were selected when the Pandora results showed that the effect of clouds was at a minimum (less minute-by-minute scatter)**

Figure 10a and b should be separate figure numbers.

**Figure 10b has been removed. A new Fig.11 has been added.**

Separate figure numbers for figure 13a and b  **Done**

Fig 13-- why are these days and pandora sites chosen? Are others worse?

**These days and sites are typical. The agreement with TCO between Pandora and OMI is very good on most days. I included a day when OMI was 3% lower than Pandora (Washington DC on 21 August) but matched EPIC more closely.**

Line 267--  No figure showing the OMI seasonal variation in TCHCHO.

**New figures show OMI TCHCHO seasonal variation**

Line 274-279-- This paragraph needs reworked. I can't tell if you are trying to say if OMI and Pandora agree on total column amounts or not. Line 276 says agreement is only good between the hours of 13-14 UTC, but what other time period would you be comparing to OMI?  **OMI comparisons are best made with Pandora at 13-14 hours local solar time.  OMI and Pandora do not agree on magnitude of TCHCHO and TCNO2 but do agree with TOC**

Line 282-- Authors only show data for 6 pandora stations in comparison with DSCOVR-EPIC. All in Eastern US and Canada and in August. Yet this statement suggests that all pandoras have good agreement. I would like to see a figure with all pandoras (or grouped by either time of year or location) before I accept that pandoras as a whole agree with DSCOVR-EPIC.

**You are right. I have not checked all 150 working locations against EPIC ozone. However, so far, I have found 6 Pandora sites with possible stray light problems Pan 63 La Porte Texas, Pan 78 Banting, Pan 73 Islamabad, Pan 260 Cameron LA, Pan 183 Londonderry New Hampshire, and Pan 77 Singapore that cannot be used for comparisons with EPIC or OMI. The ozone comparisons were included to show that both OMI and Pandora instrument calibrations are correct since total column ozone changes slowly with distance making the spatially coarse OMI retrievals comparable to Pandora, unlike NO$_2$ and HCHO.**

Technical Corrections:

Line 148—typo    **Fixed**

Line 239-- typo on figure number    **Fixed**

Reply    **Citation**: https://doi.org/10.5194/egusphere-2024-1216-RC

---

## Referee Report (RR1)

The paper by Herman and Mao is a study comparing Total Column HCHO, NO2, and O3 from Pandora Spectrometers to OMI and DSCOVER-EPIC. They included multiple pandora stations located at various locations around the globe and during different seasons. They found that agreement is overall good, however OMI does not always capture the seasonal variation as seen in the pandoras and may not be sensitive to changes in surface concentrations. DSCOVER-EPIC agrees quite well with the diurnal pandora data. This is a much needed comparison study as there are few publications on the validity of pandora spectrometers which are to be used in future satellite validation plans. The authors have addressed some of the previous comments, but not all. Below I note additional questions or comments based on this revised version.

Larger suggestions include a more in depth discussion on the Pandora's data quality. The authors use numerous Pandoras around the globe in different figures without providing a reason for the change. Keeping the study limited to a couple would provide a clearer conclusion. The authors also emphasize that there are disagreements between OMI and Pandora without much explanation as to the cause.

Line 22: switch 'OMI' and 'Ozone Monitoring Instrument'. OMI is the abbreviation.

Line 24: put 'TCHCHO' and 'TCNO2' in parenthesis

Line 98: I still don't understand why you're not using TROPOMI at all but if you're not using it, remove the mention of it here. Not relevant.

Line 100: I still would like to see more explanation of the data filtering in the text. There is only a brief mention on line 145 of the rms. Are you not considering the independent uncertainty, negative values, 'unusable' data, L2 DQ flags, etc.?

Lines 132-126: I still don't think this level of detail is necessary for this publication. This info is needed for a user manual, not an intercomparison paper.

Line 139: 50 km is quite far for these Pandoras. Especially in an area such as NYC where NO2 changes.

Figure 1: Note the uncertainties for HCHO and NO2.

Figure 1: I still don't see the need for both Figure 1 and 2. They both are saying that there is a seasonal dependence at the Bronx. If you want to remove the noise from figure 1, that's fine but then I don't feel figure 1 is necessary. Other than one sentence in line 145, the weekly data in the second and third columns from figure 1 are not discussed.

Line 155: Is the HCHO seasonal dependence due direct emissions from a park in the Bronx? Not isoprene emissions that break down into HCHO?

Figure 7. Upper panels: Better label for legend. Should have 'Pandora' somewhere. Why is 'NO2 OMI' in magenta?

Line 222: I still have an issue with this sentence. You are only able to compare Pandora at the OMI overpass time. There is no point in comparing OMI to the entire diurnal data of Pandora.

Line 245: Why only restrict the cases to the 3 days shown? Why not find the agreement for the entire record?

Figure 12: The chosen Pandoras jump around too much. I would like each figure to be more consistent, so we are talking about the same location/environment for the entire paper. We can look at different areas but be more consistent. For example, figure 9 we are discussing NYC, South Korea and PA. Figure 10/11 we jump to Toronto, and figure 12 we go to Rome, PA, and CO. I would prefer to see the Lowess lines in Figure 12 of the previously discussed pandoras instead.

Line 327: What influences the agreement? Clouds? Different pandoras? Seasonal dependence. Where was this discussion in the rest of the text?

---

## Author Response (AR2)

Reviewer #1 Second review

Before publication, the authors should address methods of filtering Pandora data quality as well as a more clear explanation for the chosen Pandoras in this study. This is a much needed comparison paper that shows diurnal trends of Total column NO2 and HCHO, but little explanation is provided as to why OMI disagrees over half the time.

**I do not know why OMI and Pandora disagree. Both use similar spectral fitting algorithms with spectrometers that have similar spectral resolution. There are differences. An important difference is that Pandora is a ground-based instrument and observes the boundary layer consistently while OMI as a satellite instrument may not. For $NO_2$ retrievals in the 410 to 440 nm region, this should not be a problem, but for HCHO retrievals in the UV where there is strong $O_3$ absorption and Rayleigh scattering, there may be difficulties. The other difference is the much larger field of view of OMI compared to Pandora, which means that OMI is averaging over a much larger area.**

The paper by Herman and Mao is a study comparing Total Column HCHO, NO2, and O3 from Pandora Spectrometers to OMI and DSCOVER-EPIC. They included multiple pandora stations located at various locations around the globe and during different seasons. They found that agreement is overall good, however OMI does not always capture the seasonal variation as seen in the pandoras and may not be sensitive to changes in surface concentrations. DSCOVER-EPIC agrees quite well with the diurnal pandora data. This is a much needed comparison study as there are few publications on the validity of pandora spectrometers which are to be used in future satellite validation plans. The authors have addressed some of the previous comments, but not all. Below I note additional questions or comments based on this revised version. Larger suggestions include a more in depth discussion on the Pandora's data quality. The authors use numerous Pandoras around the globe in different figures without providing a reason for the change. Keeping the study limited to a couple would provide a clearer conclusion. The authors also emphasize that there are disagreements between OMI and Pandora without much explanation as to the cause.

Line 22: switch 'OMI' and 'Ozone Monitoring Instrument'. OMI is the abbreviation. **Done**

Line 24: put 'TCHCHO' and 'TCNO2' in parenthesis **Done**

Line 98: I still don't understand why you're not using TROPOMI at all but if you're not using it, remove the mention of it here. Not relevant. **Removed**

Line 100: I still would like to see more explanation of the data filtering in the text. There is only a brief mention on line 145 of the rms. Are you not considering the independent uncertainty, negative values, 'unusable' data, L2 DQ flags, etc.?

**time series use all available Pandora data between 07:00 and 17:00 filtered for data quality (values with large RMS errors and with negative values are removed).**

Lines 132-126: I still don't think this level of detail is necessary for this publication. This info is needed for a user manual, not an intercomparison paper. **I agree, but the poor file naming notation causes confusion for a reader trying to understand this paper.**

Line 139: 50 km is quite far for these Pandoras. Especially in an area such as NYC where NO2 changes. **The OMI data used is a $0.25^O$ x $0.25^O$ gridded data set that corresponds to about 30 x 30 $km^2$ for midlatitudes. Most of the OMI data are less than 50 km away from the Pandora location.**

Figure 1: Note the uncertainties for HCHO and NO2.  Figure 1: I still don't see the need for both Figure 1 and 2. They both are saying that there is a seasonal dependence at the Bronx. If you want to remove the noise from figure 1, that's fine but then I don't feel figure 1 is necessary. Other than one sentence in line 145, the weekly data in the second and third columns from figure 1 are not discussed.

**Figure 1 contains Pandora data between 07:00 and 17:00 local time, whereas Figure 2 contains data near the OMI overpass time showing that the seasonal dependence should be seen by OMI.**

Line 155: Is the HCHO seasonal dependence due direct emissions from a park in the Bronx? Not isoprene emissions that break down into HCHO?  Figure 7. Upper panels: Better label for legend. Should have 'Pandora' somewhere. Why is 'NO2 OMI' in magenta?

**I have modified the sentence concerning the park in the Bronx**

**The primary emission sources of atmospheric HCHO include direct emissions of HCHO precursors from vegetation and lakes, primarily through the release of biogenic volatile organic compounds such as isoprene and terpenes from vegetation….**

Line 222: I still have an issue with this sentence. You are only able to compare Pandora at the OMI overpass time. There is no point in comparing OMI to the entire diurnal data of Pandora.

**The sentence has been removed**

 Line 245: Why only restrict the cases to the 3 days shown? Why not find the agreement for the entire record?

 **I was showing examples if the diurnal variation observed by Pandora with comparosons to the OMI values. Other graphs show the long-term comparisons. The text says that Figure 8 contains typical examples of highly variable $NO_2$ variation during the day. The preceding figure 7 shows the longer term offset between Pandora and OMI for two of the three sites (Bronx and Busan) in Figure 8.**

Figure 12: The chosen Pandoras jump around too much. I would like each figure to be more consistent, so we are talking about the same location/environment for the entire paper. We can look at different areas but be more consistent. For example, figure 9 we are discussing NYC, South Korea and PA. Figure 10/11 we jump to Toronto, and figure 12 we go to Rome, PA, and CO. I would prefer to see the Lowess lines in Figure 12 of the previously discussed pandoras instead.

**Figure 10 was chosen to show a site where the Pandora is located somewhat away from a nearby city. Here Pandora 145 sees the HCHO seasonal variation that is only seen weakly by the large OMI gridded pixel. Also, for NO$_2$, OMI is seeing the emissions from Toronto, whereas Pandora located on the outskirts does not see the seasonal variation. Figure 11 shows more detail for the Toronto SC and is restricted to 13:00 to 14:00 Pandora data. Figure 12 is intended as a sampling from different regions, US, Japan, and Europe.**

Line 327: What influences the agreement? Clouds? Different pandoras? Seasonal dependence. Where was this discussion in the rest of the text?

**Ozone agreement between Pandora, OMI and EPIC is better than that for NO2 and HCHO because the large majority of TCO is in the stratosphere and changes slowly over relatively large distances in most regions. Exceptions are in mountain areas where weather driven atmospheric pressure changes can cause TCO to change rapidly over short distances. The same is true when weather fronts pass through any local area.  See line 296.**

**Additional Comments**

**Because there are now about 150 Pandora instruments operating worldwide. Of those, 75 have relatively long and complete data sets (O3, NO2, and HCHO) available. Of those, I  arbitrarily chose a subset of 30 Pandoras as given in Table 1 that covered a moderate latitude and longitude range. When comparing individual days, I selected days that were mostly cloud free as determined from the Pandora data. However, cloud-free as seen by Pandora is not necessarily completely cloud-free as seen by OMI because of OMI's much larger field of view. When there are clouds observed by Pandora the scatter in the successive data points increases. I selected most days that had little data scattering. See lines 142 and 143.**

**All of the 30 Pandoras in Table 1 have ozone values that agree with OMI. This suggests that the laboratory calibrations of the selected Pandoras between 317.5 and 388 nm are consistent and valid. The same Pandora calibration method was used for the visible range 400 to 525 nm that is needed for NO$_2$ retrievals.**

**I have added some comments in the text and the Summary.**

---

## Author Response (AR3)

**Public justification (visible to the public if the article is accepted and published):**
I agree with referee#1 that the discrepancies between OMI and Pandora NO2 and HCHO data deserves a better discussion. To my point of view, the current response lacks substance and should be enhanced with:

(1) a better review of the possible sources of bias, on both satellite and ground-based data sets. Beyond the issue of the different FOVs (that mainly affect NO2 since one expects the HCHO horizontal distribution to be more homogeneous than the NO2 one), satellite data can possibly be biased due to a number of retrieval issues, some of them affecting slant columns, others airmass factor calculations.

**This paper does not examine the horizontal distribution of either TCNO2 or TCHCHO to see how their spatial distribution may cause the Pandora results to differ from those measured by OMI. Frequently TCNO2(Pan) > TCNO2(OMI) and TCHCHO(Pan) < TCHCHO(OMI). Aside from the effect of spatial averaging by OMI compared to Pandora the effective Airmass factor for OMI retrievals is a large source of error compared to the geometric Airmass factor for Pandora direct sun measurements. The following paragraph has been added, which also includes additional references.**

Previous validation studies of $TCNO_2$ and TCHCHO have been made with emphasis on the amount of bias between ground-based and satellite retrievals of total column $NO_2$ and HCHO (Pinardi et al., 2020; de Smedt et al., 2021) and references therein. Validation studies using Pandora measurements have shown that OMI TCNO2 retrievals tend to underestimate the degree of $NO_2$ pollution, especially in urban areas where the coarse OMI spatial resolution tends to reduce the spatially averaged amount (Celarier et a., 2008; Lamsal et al., 2014; Judd et al., 2019; Zhao et al., 2019). In addition to the different field of view, the agreement between OMI and Pandora depends strongly on determining the OMI effective air mass factor for a wide variety of observing and solar zenith angles (Lorente et al., 2017), whereas Pandora uses a simple geometric direct sun airmass factor (Herman et al., 2009, Eq 3). Studies of TCHCHO involving Pandora prior to 2020 are probably not valid because of a problem with internal generation of HCHO in the Pandora instrument (Spinei et al., 2021). More recent studies (Wang et al. 2022) obtain a seasonal dependence of surface concentrations similar to the TCHCHO in this study. The largest sources of error in TCHCHO retrievals are the determinations of the air mass factor for satellite observations and the fact that ozone and formaldehyde have overlapping absorption spectra so that a small error in ozone retrieval can affect the formaldehyde results. A comparison of direct-sun Pandora TCHCHO retrievals with Geostationary Environment Monitoring Spectrometer GEMS shows a similar seasonal dependence (Fu et al., 2025).

(2) a better review of the existing litterature on satellite NO2 and HCHO validation. For NO2, there a several relevant papers, see e.g. https://amt.copernicus.org/articles/13/6141/2020/

, and references therein. For HCHO, see e.g. https://www.atmos-meas-tech.net/13/3751/2020/, and https://acp.copernicus.org/articles/21/12561/2021/

**See the above paragraph**

. How do the results presented here, compare with already published validation results?

**The seasonal behavior is similar in that HCHO usually peaks in the summer and NO2 in the winter at mid latitudes.**